# A new genomic framework to categorize pediatric acute myeloid leukemia

Masayuki Umeda[1,10], Jing Ma[1,10], Tamara Westover [1], Yonghui Ni[2], Guangchun Song[1], Jamie L. Maciaszek[1], Michael Rusch [3], Delaram Rahbarinia[3], Scott Foy[3], Benjamin J. Huang [4], Michael P. Walsh[1], Priyadarshini Kumar[1], Yanling Liu[3], Wenjian Yang [5], Yiping Fan[6], Gang Wu [1,6], Sharyn D. Baker[7], Xiaotu Ma [3], Lu Wang [1], Todd A. Alonzo[8], Jeffrey E. Rubnitz [9], Stanley Pounds [2] & Jeffery M. Klco [1] ✉

Recent studies on pediatric acute myeloid leukemia (pAML) have revealed pediatric-specific driver alterations, many of which are underrepresented in the current classification schemas. To comprehensively define the genomic landscape of pAML, we systematically categorized 887 pAML into 23 mutually distinct molecular categories, including new major entities such as *UBTF* or *BCL11B*, covering 91.4% of the cohort. These molecular categories were associated with unique expression profiles and mutational patterns. For instance, molecular categories characterized by specific *HOXA* or *HOXB* expression signatures showed distinct mutation patterns of *RAS* pathway genes, *FLT3* or *WT1*, suggesting shared biological mechanisms. We show that molecular categories were strongly associated with clinical outcomes using two independent cohorts, leading to the establishment of a new prognostic framework for pAML based on these updated molecular categories and minimal residual disease. Together, this comprehensive diagnostic and prognostic framework forms the basis for future classification of pAML and treatment strategies.

Acute myeloid leukemia (AML) is characterized by aberrant clonal expansion of hematopoietic progenitors with differentiation defects[1–3]. Although pAML shares many clinical and pathological characteristics with adult AML, genetic differences have also been appreciated[4,5]. Notably, t(11;x), resulting in *KMT2A* rearrangements, are more common in pAML, and adult AML frequently harbors mutations in *DNMT3A* and splicing factor genes, whereas core binding factor (CBF) AMLs are common across the age spectrum[4]. In addition, progress in diagnostic technologies has identified cryptic fusions of *NUP98* (ref. 6) and

*GLIS* family[7] members and *UBTF* tandem duplications[8] enriched in pAML. Recent updates in the World Health Organization classification[9] (WHO[5th]) and the International Consensus Classification[10] (ICC) define AMLs with *KMT2A* and *NUP98* rearrangements as distinct disease entities. However, recently discovered recurrent driver alterations in pAML remain categorized as 'acute myeloid leukemia with other defined genetic alterations' or 'AML, not otherwise specified (NOS)', confirming the need to understand both the biological and clinical features of pAMLs with these driver alterations.

[1]Department of Pathology, St. Jude Children's Research Hospital, Memphis, TN, USA. [2]Department of Biostatistics, St. Jude Children's Research Hospital, Memphis, TN, USA. [3]Department of Computational Biology, St. Jude Children's Research Hospital, Memphis, TN, USA. [4]Department of Pediatrics, University of California San Francisco, San Francisco, CA, USA. [5]Department of Pharmacy and Pharmaceutical Sciences, St. Jude Children's Research Hospital, Memphis, TN, USA. [6]Center for Applied Bioinformatics, St. Jude Children's Research Hospital, Memphis, TN, USA. [7]Division of Pharmaceutics and Pharmacology, College of Pharmacy, Comprehensive Cancer Center, The Ohio State University, Columbus, OH, USA. [8]Department of Population and Public Health Sciences, Keck School of Medicine, University of Southern California, Los Angeles, CA, USA. [9]Department of Oncology, St. Jude Children's Research Hospital, Memphis, TN, USA. [10]These authors contributed equally: Masayuki Umeda, Jing Ma. ✉e-mail: jeffery.klco@stjude.org

Accumulation of clinical outcomes associated with gene alterations enabled risk stratification of adult AML according to detailed mutational profiling, such as the 2022 European LeukemiaNet risk stratification[11]. By contrast, risk stratification for pAML is still developing, and various strategies are utilized in clinical trials[12–15]. This is partly due to genetic differences between adult and pAML[4], the rarity of the disease and a shortage of clinical outcome studies related to genetic alterations. To clarify the genomic landscape of pAML and its association with clinical outcomes, we characterized 887 cases of pAML by transcriptome and genome profiling. These analyses resulted in 23 molecular categories, defined by mutually exclusive gene alterations and specific expression profiles that show unique biological and mutational characteristics. These molecular categories have predictive value regarding clinical outcomes that can be leveraged to establish a framework for diagnosis and outcome prediction.

## Results

### Comprehensive genetic characterization of pAML

pAML samples were collected from previously published studies[4,7,8,16–25] or at St. Jude Children's Research Hospital, resulting in a cohort of 887 unique pAMLs either at diagnosis ($n = 783$, 88.3%) or at relapse ($n = 104$, 11.7%) (Fig. 1a, Extended Data Fig. 1a and Supplementary Table 1). This pAML cohort showed a wide age distribution at diagnosis (range 0–23.5 years; median 9.3) including young adults, with peaks in infancy and adolescence (Extended Data Fig. 1b). We first assessed the genetic landscape of these AMLs using RNA sequencing (RNA-seq) data to detect fusions, internal or partial tandem duplications (ITD/PTD), copy-number variants (CNV), as well as single nucleotide variants (SNV) and insertions and deletions (indels) (Fig. 1a–e, Extended Data Fig. 1c–e and Supplementary Tables 2–9). For 665 cases (74.9%) with either whole-genome sequencing (WGS, 59.2%) or whole-exome sequencing (WES, 44.0%), we also collected processed data from publications or performed de novo calling for newly included samples, which validated 97.3% of calls from the RNA-seq pipeline[8] (Fig. 1a and Extended Data Fig. 1f).

Pathogenic fusions or structural variants (SVs) were identified in 627 patients (70.7%). Most of these are recurrent and class-defining in pAML (for example, KMT2Ar, 20.3%; RUNX1::RUNX1T1, 12.4%) (Fig. 1b and Supplementary Table 6), whereas we also found fusions recurrent in other leukemias, such as SET::NUP214 ($n = 1$) or SFPQ::ZFP36L2 ($n = 1$). Mutational profiling revealed 1,924 pathogenic or likely pathogenic somatic mutations in 749 (84.4%) patients, including class-defining NPM1 (67 patients, 7.6%) and CEBPA (49 patients, 5.5%) mutations (Fig. 1c and Supplementary Tables 7 and 8). Most mutations were in genes involved in signaling pathways ($n = 865$), epigenetics ($n = 312$) and transcription factors ($n = 432$). RAS pathway mutations were most frequent, with 37.5% (333 of 887) having at least one RAS-related mutation and 21.3% of those (71 of 333) having mutations in multiple RAS pathway genes. Gains of chromosome 8 (7.3%) or chromosome 21 (6.2%) and loss of the long arm of chromosome 5 (5q-: 1.5%) or chromosome 7 (4.8%) were commonly observed (Fig. 1d, Extended Data Fig. 1e and Supplementary Table 9). Enrichment of focal deletions involving RB1

(13q14: 2.9%), ETV6 (12p13: 2.1%), NF1 (17q11: 2.0%) and TP53 (17p13: 2.0%), and focal gains involving AKT3 and FH (1q43: 3.0%) or ABCA transporters (17q24: 1.9%) were also identified. Genomic random interval (GRIN) analysis[26] identified 142 altered genes with statistical significance (Fig. 1e and Supplementary Table 10). Consistent with previous reports, RAS-related mutations or FLT3-ITD with variable variant allele frequencies (VAFs) were highly co-occurring with class-defining alterations (Fig. 1e and Extended Data Fig. 2a,b). By contrast, mutations in UBTF or CBFB were exclusively found in cases without a defining alteration, as previously shown[8,27], suggesting that these alterations define subgroups with distinct molecular characteristics.

Based on these collective data, we classified pAMLs using current WHO and ICC systems, and the frequencies of major classifications are consistent with cytogenetic profiles of European pAML cohorts[28,29] (Fig. 1e,f, Extended Data Fig. 1g and Supplementary Fig. 1). In our pAML cohort, 68.5% of cases had specified genetic alterations in WHO[5th], 10.7% of cases were defined as 'acute myeloid leukemia, myelodysplasia-related' (AML-MR) and the remaining cases with rare fusions or no defining alteration were classified as 'acute myeloid leukemia with other defined genetic alterations' (15.8%) or by differentiation stages (3.4%). By contrast, 95.0% of adult AMLs can be classified either by specific gene alteration (67.1%) or as AML-MR (27.8%)[30], emphasizing the need for a more comprehensive classification of pAML based on its unique biology.

### Molecular categories defined by unique gene alterations

We and others have shown that class-defining driver alterations are associated with specific expression patterns[8,31] or that allele-specific and outlier expression of MECOM[32,33], BCL11B (ref. 34) or MNX1 (ref. 35) by SVs can define subtypes. We then integrated the mutational landscape with expression profiling to define granular molecular categories for pAML (Supplementary Table 11). Uniform manifold approximation and projection (UMAP) analysis of transcriptional data revealed tight clustering of classes defined in WHO[5th], including RUNX1::RUNX1T1, CBFB::MYH11 and CEBPA mutation, suggesting subtype-specific expression patterns (Fig. 2a and Extended Data Fig. 3a). We noted that clustering is also driven in part by differentiation status represented by marker gene expression, French–American–British (FAB) classification or cellular hierarchy[36] (Extended Data Fig. 3c–e), contributing to heterogeneity within large categories such as KMT2Ar or NUP98r (Fig. 2a and Extended Data Figs. 3a and 4a). Diffusion maps[37] confirmed similar patterns of clustering and differentiation status (Extended Data Fig. 3a–e). Cases with NPM1 fusions or indels outside the C terminus[38] clustered with canonical NPM1 mutations, and we assigned them to the NPM1 category (Extended Data Fig. 4a); similarly, we assigned a RAR family fusion, TBL1XR1::RARB, to the acute promyelocytic leukemia category based on expression similarities with PML::RARA cases. Among the remaining cases without class-defining alterations, we found that the following alterations were also mutually exclusive and thus defined them as independent molecular categories: UBTF tandem duplications[8], GLIS family (GLIS2-3) fusions[7], fusions of FET and ETS family genes[39,40] (for example, FUS::ERG), BCL11B SVs[34] (Supplementary Table 12),

**Fig. 1 | Comprehensive genetic characterization of pAML. a**, Study cohort of pAML ($n = 887$) and study design. **b**, Recurrent pathogenic or likely pathogenic in-frame fusions (blue) and SVs (gray) detected in the entire cohort ($n \geq 3$). Fusions included only in-frame fusions, and SVs included out-of-frame fusions resulting in loss of the C terminus of the protein and alterations detected from WGS data using CREST. **c**, Recurrent pathogenic or likely pathogenic somatic mutations ($n \geq 15$). Colors represent types of mutations. Bars in **b** and **c** represent the total number of alterations in the cohort. **d**, Results of GISTIC analysis for focal chromosomal events (shorter than 90% of the chromosome arm). The left-hand panel shows the enrichment of focal gains (red) and the right-hand panel shows the enrichment of focal losses (blue). Green lines show a significance threshold for $q$ values (0.25). Representative genes in enriched regions are highlighted. **e**, Genomic landscape and WHO classification of pAML. Representative genes from GRIN analysis or defining alterations are shown. Colors represent types of mutations. Numbers of gene alterations are shown next to gene names, and the lines of the box plot for VAFs represent the 25% quantile, median and 75% quantile. The upper whisker represents the higher value of maxima or 1.5× the interquartile range (i.q.r.) and the lower whisker represents the lower value of minima or 1.5× i.q.r. **f**, Summary of the WHO classification (WHO[5th]) of the entire cohort. A box with solid lines indicates categories with defining driver alterations. Boxes with dashed lines indicate subgroups with specified gene alterations, myelodysplasia-related or other defined genetic alterations. NA, Not available.

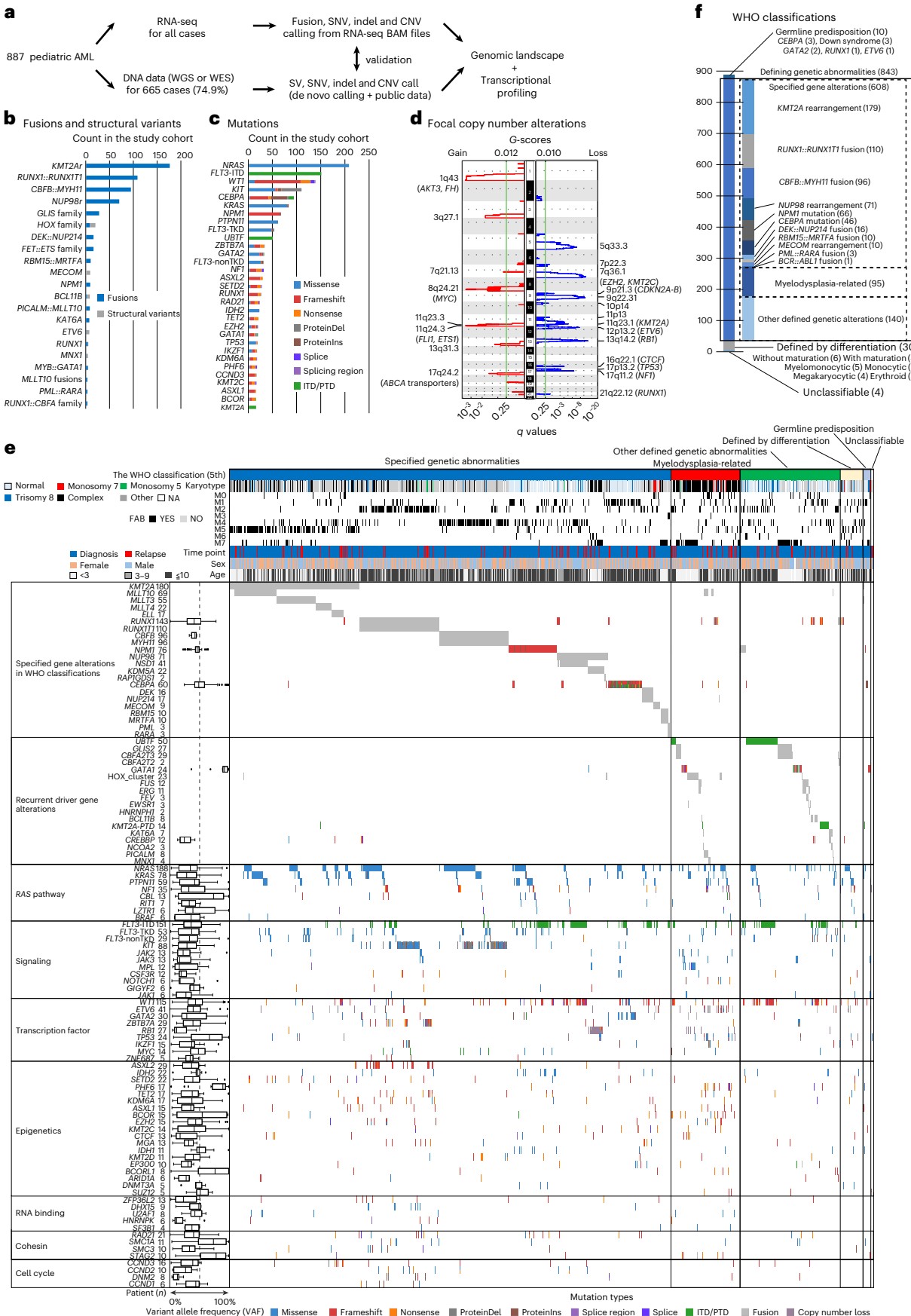

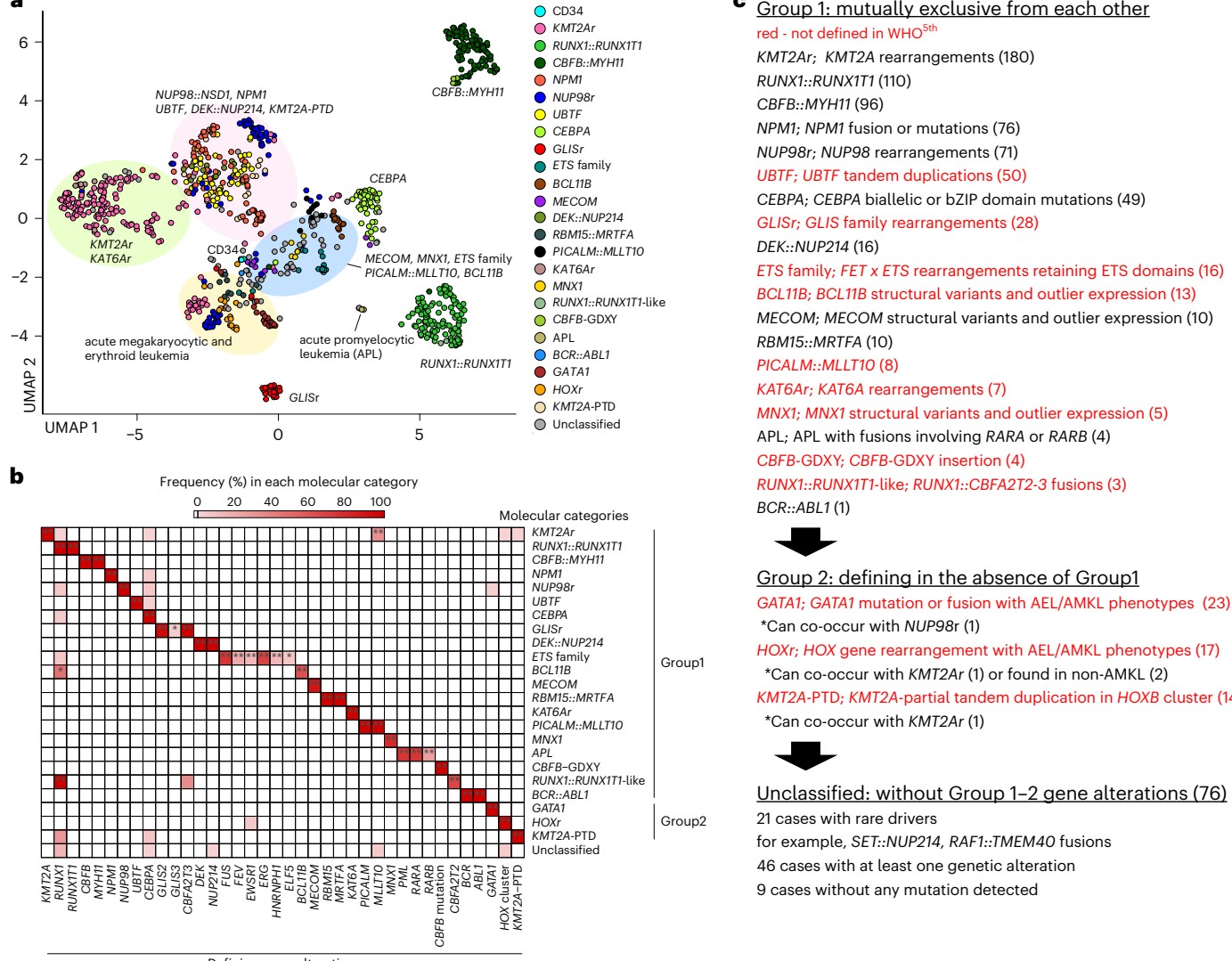

**Fig. 2 | Molecular categories defined by mutually exclusive gene alterations.**
**a**, UMAP plot of the entire pAML cohort (*n* = 887) and cord blood CD34⁺ cells (normal controls: *n* = 5) using the top 315 variable genes. The colors of each dot denote the molecular categories of the samples. Representative category names are shown, and large clusters enriching specific categories are highlighted in circles (pink: *NUP98::NSD1, NPM1, UBTF, DEK::NUP214, KMT2A*-PTD; green: *KMT2A*r and *KAT6A*r; yellow: categories with acute megakaryocytic or erythrocytic expression; blue: *MECOM, MNX1, ETS* family, *PICALM::MLLT10, BCL11B*).
**b**, Heatmap showing frequencies of defining gene alterations represented by color. Statistical significance was assessed by two-sided Fisher's exact test to calculate *P* values of co-occurrence, followed by Benjamini–Hochberg adjustment for multiple testing to calculate *q* values (**P* < 0.05, ***q* < 0.05). **c**, Definition of molecular categories and diagnostic workflow. Molecular categories not defined in WHO[5th] are highlighted in red. APL, acute promyelocytic leukemia.

*PICALM::MLLT10, KAT6A* rearrangements, *MNX1* SVs[41], *RUNX1* fusion with *CBFA2T2-3* (ref. 42) (*RUNX1::RUNX1T1*-like) and newly reported *CBFB* insertions (*CBFB*-GDXY)[27] (Fig. 2a–c). *GATA1* fusions (for example, *MYB::GATA1*) or mutations involving *HOX* cluster genes and *KMT2A*-PTD could rarely co-occur with the above-mentioned category-defining alterations (Fig. 2b). However, they were still predominantly found in cases without category-defining alterations and assigned to these categories only with consistent expression patterns and without previously explained driver alterations. By contrast, defining mutations of AML-MR in WHO[5th] were overall rare (range 0.1–2.1%), frequently co-occurred with other defining alterations (for example, *EZH2* in *PICALM::MLLT10*), and could be found in various clusters rather than as a distinct group (Extended Data Fig. 3a,f), leading to its exclusion as a defining category for pAML. In addition to 11 categories defined by WHO[5th], this pAML classification system with 12 new molecular categories captures 91.4% of pAML cases, contrasting to 68.5% by WHO[5th] (Fig. 3).

## Biological characterization of the molecular categories

Establishing updated molecular categories for pAML allowed for the investigation of clinicopathological associations. Categories with acute megakaryoblastic leukemia (AMKL) or acute erythroid leukemia (AEL) phenotypes are clearly enriched in infants, whereas CBF leukemias and mutation-defined leukemias (for example, *UBTF*, *NPM1*, *CEBPA*) were enriched in adolescents and young adults (Fig. 4a and Extended Data Fig. 4b). Notably, among *KMT2A* fusion partners, *MLLT3* and *MLLT10* were found in both monocytic AML and AMKL; however, these fusions preferentially show AMKL phenotypes in infants, suggesting that AMKL phenotypes are defined both by driver alterations and by developmental stages as discussed previously[43,44]. Overall, however, each molecular category showed variable morphological features represented by FAB classification, except categories with acute promyelocytic leukemia (M3) or AMKL (M7) phenotypes. Likewise, complex karyotypes, which also define AML-MR[9], were frequently observed in *MNX1, HOX*r and *PICALM::MLLT10* categories. In

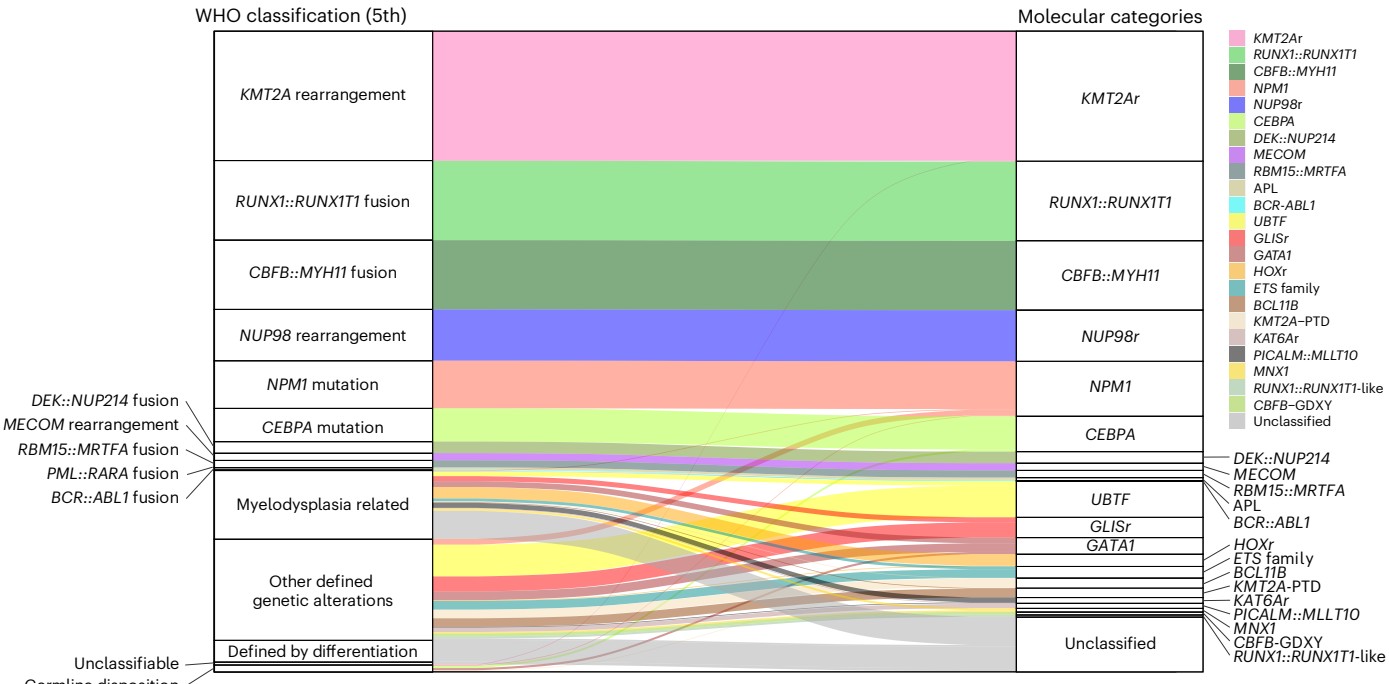

**Fig. 3 | Comparison between molecular categories and the WHO classification.** The colors of the ribbon plot represent molecular categories of samples in the pAML cohort ($n$ = 887).

addition, many of these category-defining alterations are cytogenetically cryptic (for example, *NUP98::NSD1* or *GLIS* family) or somatic mutations (for example, *CEBPA*, *UBTF* or *GATA1*), highlighting the need for sequencing approaches for the appropriate molecular diagnosis of pAML.

We next explored the association between defining alterations and cooperating mutations, because some cooperating mutations co-occur and act synergistically with specific driver events[4,45]. Signaling alterations were broadly found in 66.3% of patients, although each mutation showed distinct patterns among molecular categories with variable VAFs (Figs. 1e and 4b). Among *RAS* mutations, *NRAS* mutations were broadly found and enriched in *CBFB::MYH11* and *NPM1*, whereas *KRAS* mutations were enriched in *KMT2A*r and *DEK::NUP214*. Similarly, *FLT3*-ITD showed strong enrichment in *NUP98*r, *NPM1*, *UBTF*, *KMT2A*-PTD and *BCL11B*, accounting for 66.2% of *FLT3*-ITD+ cases, whereas 75.5% of *FLT3*-TKD (tyrosine kinase domain) were found in *KMT2A*r, *NPM1* and CBF-AMLs. Similarly, *WT1* mutations were specifically enriched in *NUP98*r, *UBTF* and *BCL11B*, and highly co-occurring with *FLT3*-ITD (Fig. 4b).

We further evaluated gene expression signatures among molecular categories. Top variably expressed genes across the cohort are involved in development, differentiation or inflammation (Extended Data Fig. 5a and Supplementary Table 13), consistent with previous reports that the heterogeneity of AML can be partly attributed to differentiation status[3,36,46]. Gene set enrichment analysis (GSEA) confirmed known expression profiles of major categories (Fig. 4c and Supplementary Table 14), whereas the new categories proposed in this study show similarities and differences with canonical categories. For example, *UBTF* showed expression signatures similar to *NPM1* and *DEK::NUP214*, whereas *KAT6A*r was similar to *KMT2A*r, suggesting shared biological mechanisms. In addition, genes involved in signaling pathways, immunity or drug resistance showed unique enrichment across categories. Weighted gene coexpression network analysis (WGCNA)[47] confirmed characteristic patterns of active gene networks associated with specific biological functions in each category (Extended Data Fig. 5b and Supplementary Table 15).

Given recent adult AML-focused studies uncovering the associations of cellular stemness[48,49] or hierarchy[36,50] with prognosis or drug response, we investigated these features in our pAML dataset. We observed unique patterns of stemness and cellular hierarchy scores in each category. Molecular categories known to have a good prognosis (*RUNX1::RUNX1T1*, *CBFB::MYH11* and *CEBPA*) tended to have high granulocyte–monocyte progenitor (GMP) scores (median >0.20) (Fig. 4d and Extended Data Fig. 5c), except for the low GMP scores (median 0.078) and mid-high stemness-related scores in *NPM1*. Also, *KMT2A*r, associated with poor prognosis, showed low stemness-related scores and variable differentiation-related scores. Also, various prognostic scores (for example, LSC17 (ref. 48), iScore[46]) correlated with molecular categories (Extended Data Fig. 5d), collectively demonstrating that molecular categories are associated with unique pathophysiological characteristics.

## Superfamilies defined by *HOX* gene expression profiles

These molecular categories also showed intercategorical similarities, forming large clusters of AMKL/AEL, immature AML, CBF leukemias, *CEBPA* and two clusters demarcated by *HOXA* and *HOXB* gene expression (Fig. 5a,b). The cluster with high *HOXA* gene expression and low *HOXB* gene expression consisted mainly of *KMT2A*r and *KAT6A*r (herein referred to as the *HOXA* group), and the other cluster characterized by high expression of both *HOXA* and *HOXB* genes included *NPM1*, *NUP98*r, *UBTF*, *KMT2A*-PTD and *DEK::NUP214* (*HOXB* group), which are generally associated with poor prognosis except for *NPM1* (Extended Data Fig. 6a). Overall, *HOXA* and *HOXB* groups, not including those with AMKL features, account for 18.5% and 23.3% of the cohort, respectively. Differential gene expression analyses revealed that *HOXB* pAMLs had high expression of stemness-related genes (*PRDM16* and *NKX2-3*) or differentiation genes (*CD96* and *WT1*) (Fig. 5c,d and Supplementary Table 16). By contrast, *HOXA* group cases showed high expression of monocyte or signaling-related genes. GRIN analysis also revealed striking differences in mutational patterns between *HOXA* and *HOXB* groups (Fig. 5e,f and Supplementary Table 17). *FLT3* was significantly altered in both groups but with different mutation types; *FLT3*-TKD

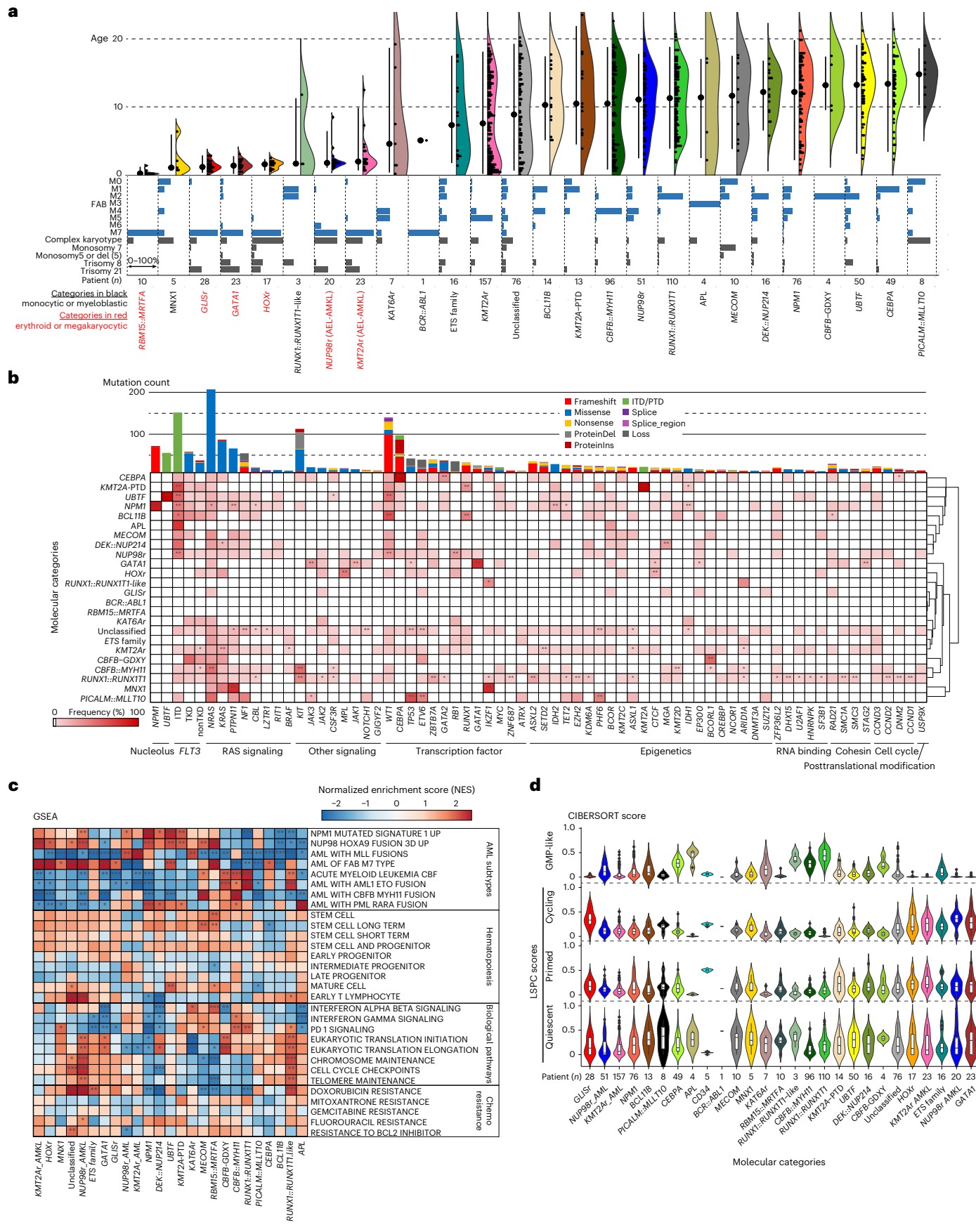

**Fig. 4 | Clinical and molecular profiles of molecular categories. a**, Clinical background of molecular categories. (Upper) Violin plots showing age distribution within each molecular category. Colors show the molecular categories. Large dots and bars represent the median and the 2.5–97.5 percentile range, respectively. Small dots represent the ages of individuals ($n$ = 887). (Lower) Frequency of FAB classification (blue bars) and karyotype (gray bars) in individual categories. **b**, Mutational heatmap showing mutation frequencies in each molecular category. The color of each panel represents the frequency of a mutation in each molecular category, and the statistical significance was assessed by two-sided Fisher's exact test to calculate $P$ values of co-occurrence followed by Benjamini–Hochberg adjustment for multiple testing to calculate $q$ values (*$P$ < 0.05, **$q$ < 0.05 after adjustment). Bars in the upper panel show the

frequency of mutations in the entire cohort, and the colors represent mutation types. Molecular categories are clustered according to Ward clustering using the Euclidean distance of the frequency matrix. Genes are grouped according to functional annotations. **c**, Heatmap showing normalized enrichment scores (NES) and FDR of GSEA for each molecular category. Colors denote NES and asterisks show FDR (*FDR < 0.05, **FDR < 0.01, ***FDR < 0.001). Detailed results are found in Supplementary Table 14. **d**, Violin plots showing cellular hierarchy scores in each molecular category inferred by CIBERSORT. The colors show molecular categories. The lines of the boxes represent the 25% quantile, median and 75% quantile. The upper whisker represents the higher value of maxima or 1.5× i.q.r., and the lower whisker represents the lower value of minima or 1.5× i.q.r. Dots show outliers. LSPC, leukemic stem and progenitor cell.

was dominant in the *HOXA* group and *FLT3*-ITD was prevalent in the *HOXB* group, accounting for 67.5% of *FLT3*-ITD+ patients (Fig. 5f and Supplementary Fig. 6b). *WT1* mutations were preferentially found in the *HOXB* group (57.6%). *FLT3*-ITD (ref. 51 and *WT1* mutations[16,52] have been associated with poor prognosis; however, these data suggest that *FLT3*-ITD and *WT1* mutations highly confound with specific driver alterations that converge on a common expression signature. *KRAS* mutations were strongly associated with the *HOXA* group and were rare in the *HOXB* group (20.7% and 3.9%, respectively). In comparison, *NRAS* mutations were prevalent in both *HOXA* and *HOXB* groups (20.7% and 17.4%) (Fig. 5f); however, *p*.G12 or *p*.G13 mutations were comparable in both categories, whereas *p*.Q61 mutations were more frequent in the *HOXA* group (Extended Data Fig. 6b). It is well-established that each *RAS* mutation has preferential distribution among cancer subtypes[53]. Expression levels or differences in the downstream signaling of RAS proteins are postulated as the possible mechanisms, and similarly, between *FLT3*-ITD and TKD[54], whereas these genes were homogenously expressed at the RNA level (Extended Data Fig. 6b). Despite varied clinical associations, these molecular category-dependent transcriptional and mutational patterns may reflect shared biology within each HOX group[55], and the different signaling dependencies may suggest targeted therapies guided by these biological insights.

Along with the global distinction between *HOXA* and *HOXB* groups, we also noted heterogeneity within each HOX cluster. The *HOXA* cluster consisted of subclusters characterized by *MECOM* or *LAMP5* expression (Extended Data Fig. 7a–c and Supplementary Table 18), harboring most *KMT2A*r cases (136 of 180; 75.6%). Notably, the largest subcluster expressed *XAGE1* family genes specifically (Extended Data Fig. 7b,c), which encode testis-specific proteins postulated as therapeutic targets in various tumors[56]. Also, the remaining *KMT2A*r cases were clustered with other categories with *HOXB* expression or AMKL less frequently. These clustering patterns were associated with age or fusion partners (for example, *KMT2A::ELL* in the *HOXB* cluster), but the associations were not exclusive (Extended Data Fig. 7d,e). Among *KMT2A*r, fusion partners and *MECOM* expression have been reported to be prognostic; however, our data suggest considerable heterogeneity in expression patterns not explained by only fusion partners or *MECOM* expression.

The *HOXB* cluster showed similar heterogeneity represented by cellular hierarchies (Extended Data Fig. 7f–h). These heterogeneities were occasionally associated with molecular categories or somatic mutations but were not exclusive, with possible factors, including cell-extrinsic factors[46] to be investigated.

### Molecular basis of AML without defining gene alterations

Seventy-six 'Unclassified' cases remained after assignment to these 23 molecular categories. Twenty-one cases had recurrent driver alterations previously reported in the literature (Fig. 6a and Supplementary Table 19), including rare in-frame *RUNX1* fusions ($n$ = 2: *USP42*; $n$ = 1: *EVX1* and *ZEB2*) and *MLLT10* fusions ($n$ = 1: *DDX3X*, *TEC* and *MAP2K2*), which require a larger cohort for further categorization. Also, in addition to high-allelic burden *JAK2 p*.V617F mutation ($n$ = 1), we found candidate driver somatic mutations of *MLLT1 p*.C119SPAR ($n$ = 1) and *H3F3A p*.K28M ($n$ = 1) in cases in HOX clusters (Fig. 6a and Extended Data Fig. 8a). These mutations resemble recurrent mutations in other pediatric cancer types with HOX gene expression and immature phenotypes (*MLLT1 p*.C118QPPG in Wilms tumor[57] or *H3F3A p*.K28M in high-grade glioma[58]), postulating a shared mechanism of tumorigenesis among these pediatric neoplasms.

Pathogenic alterations were not identified in 9 of the remaining 55 Unclassified cases, partly attributed to the lack of WGS data for 8 of these cases. The rest had at least one pathogenic, but not subtype-defining alteration enriched in *ETV6*, *RUNX1*, *TP53* and myelodysplasia-related genes in addition to complex karyotypes or monosomy 7 (Fig. 6b,c and Supplementary Tables 19 and 20). Of note, AML-MR defining karyotypes (complex karyotypes or monosomy 7) or somatic mutations were found broadly in various clusters (Extended Data Fig. 8b–d), suggesting that these alterations do not define specific categories. By contrast, *ETV6* and *RUNX1* alterations not defining established categories were found preferentially in clusters associated with FAB M0/1 or immature or T cell-like signatures (Fig. 6d, Extended Data Fig. 8b–d and Supplementary Table 21), as previously described[59]. Although various *ETV6* or *RUNX1* alterations can be class-defining (for example, *RUNX1::RUNX1T1*) or co-occur with other defining alterations, those in the Unclassified category are commonly loss-of-function

**Fig. 5 | Categories demarcated by *HOXA* and *HOXB* cluster expression. a**, UMAP plot colored according to groups of molecular categories based on UMAP clustering and *HOX* cluster gene expression profiles. A gray circle indicates a cluster enriching categories with immature phenotypes (*BCL11B, MECOM, MNX1, PICALM::MLLT10, ETS* family). **b**, *HOXA9* and *HOXB5* expression on UMAP plots. Dot colors represent the relative expression of the genes. **c**, Volcano plot showing differentially expressed genes between *HOXA* and *HOXB* groups. Genes with absolute fold change >2 and FDR < 0.05 are considered differentially expressed. Red or blue dots show genes enriched only in either *HOXA* or *HOXB* groups, respectively, and representative gene names are shown. **d**, Gene Ontology term analyses of genes with significantly high expression in *HOXA* (red) and *HOXB* (blue) categories by DAVID (Database for Annotation, Visualization and Integrated Discovery). Bars represent logged FDR. **e**, Plots showing the

results of GRIN analyses in the *HOXA* group (horizontal axis) and *HOXB* group (vertical axis). Genes with FDR < 0.05 in either the *HOXA* or *HOXB* group are shown. Red or blue dots show genes enriched only in either the *HOXA* or *HOXB* group, respectively. Dotted lines represent thresholds for statistical significance (FDR < 0.05). **f**, Mutational heatmap comparing patterns between the *HOXA* and *HOXB* groups. Colors represent mutation types, and molecular categories are annotated on the top. Bar plots on the right show frequencies of mutations in the *HOXA* and *HOXB* groups. Statistical significance of GRIN analysis in the *HOXA* and *HOXB* groups (*FDR < 0.05) and two-sided Fisher's exact test between *HOXA* and *HOXB* groups (*$P$ < 0.05, **$q$ < 0.05 after Benjamini–Hochberg adjustment) are also shown. GRIN results for *FLT3* are for the entire gene, whereas Fisher's tests were performed separately for ITD, TKD and non-TKD mutations.

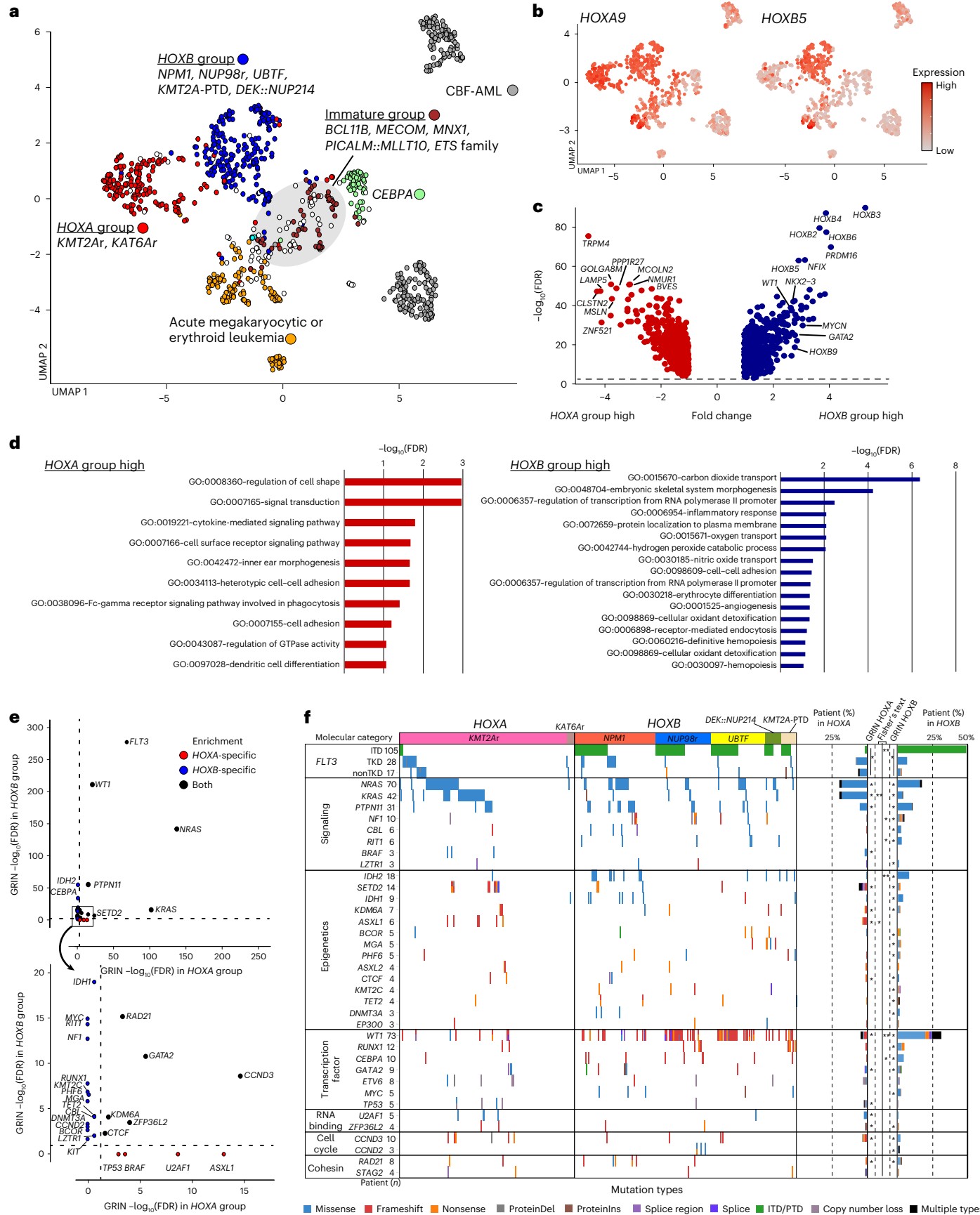

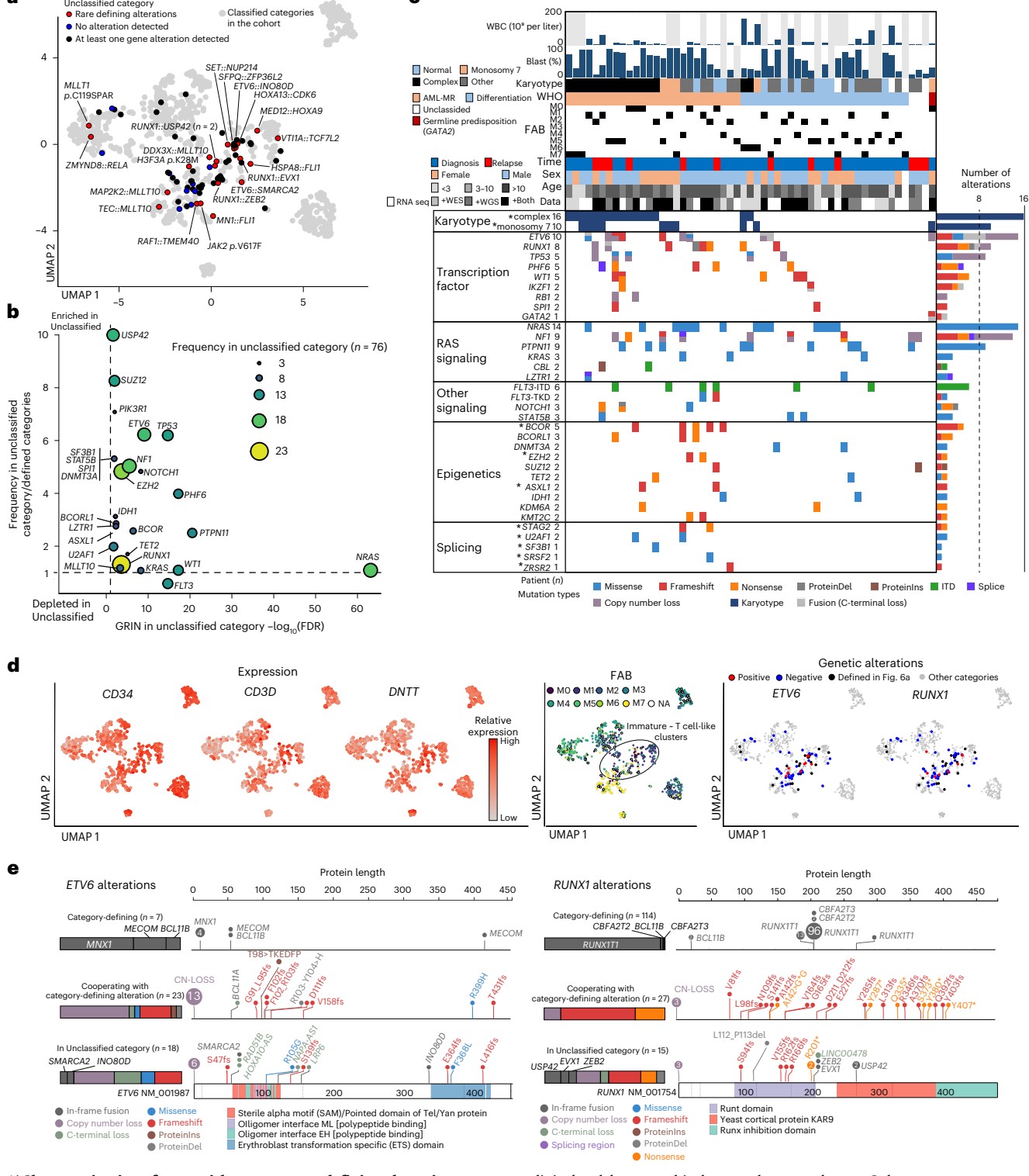

**Fig. 6 | Characterization of cases without category-defining alterations.**
**a**, UMAP plot showing cases without category-defining alterations. Red dots
represent cases with rare recurrent gene alterations, blue dots represent cases
for which no pathogenic alteration was found and black dots represent cases
with at least one gene alteration not defining the phenotype. Gray dots represent
cases with classified categories. **b**, Plot showing the FDR of GRIN analysis for the
Unclassified category (horizontal axis) and relative enrichment of the alteration
in the unclassified category (vertical axis). Dot sizes and colors denote the
Unclassified category's frequency, which included fusions, mutations, copy-
number loss and gain, and copy-neutral heterozygosity. **c**, Mutational heatmap of
the Unclassified cases, including complex karyotypes and monosomy 7. Patients'

clinical and demographic data are shown on the top. Colors represent
mutation types. Defining alterations for AML-MR are marked by asterisks.
**d**, UMAP plots showing *CD34*, *CD3D* and *DNTT* expression (left), FAB
classification (middle) and cases with *ETV6* alterations and *RUNX1* alteration
(right). For *ETV6* and *RUNX1* alteration plots, cases with classified categories
are shown as gray dots. **e**, Patterns of alteration in *ETV6* (left) and *RUNX1* (right).
Category-defining fusions are shown in the top row, alterations co-occurring
with category-defining alterations in the middle row, and alterations in the
Unclassified category in the bottom row. Bars represent a relative fraction of
alteration in each group and colors denote the alteration types. WBC, white
blood cell.

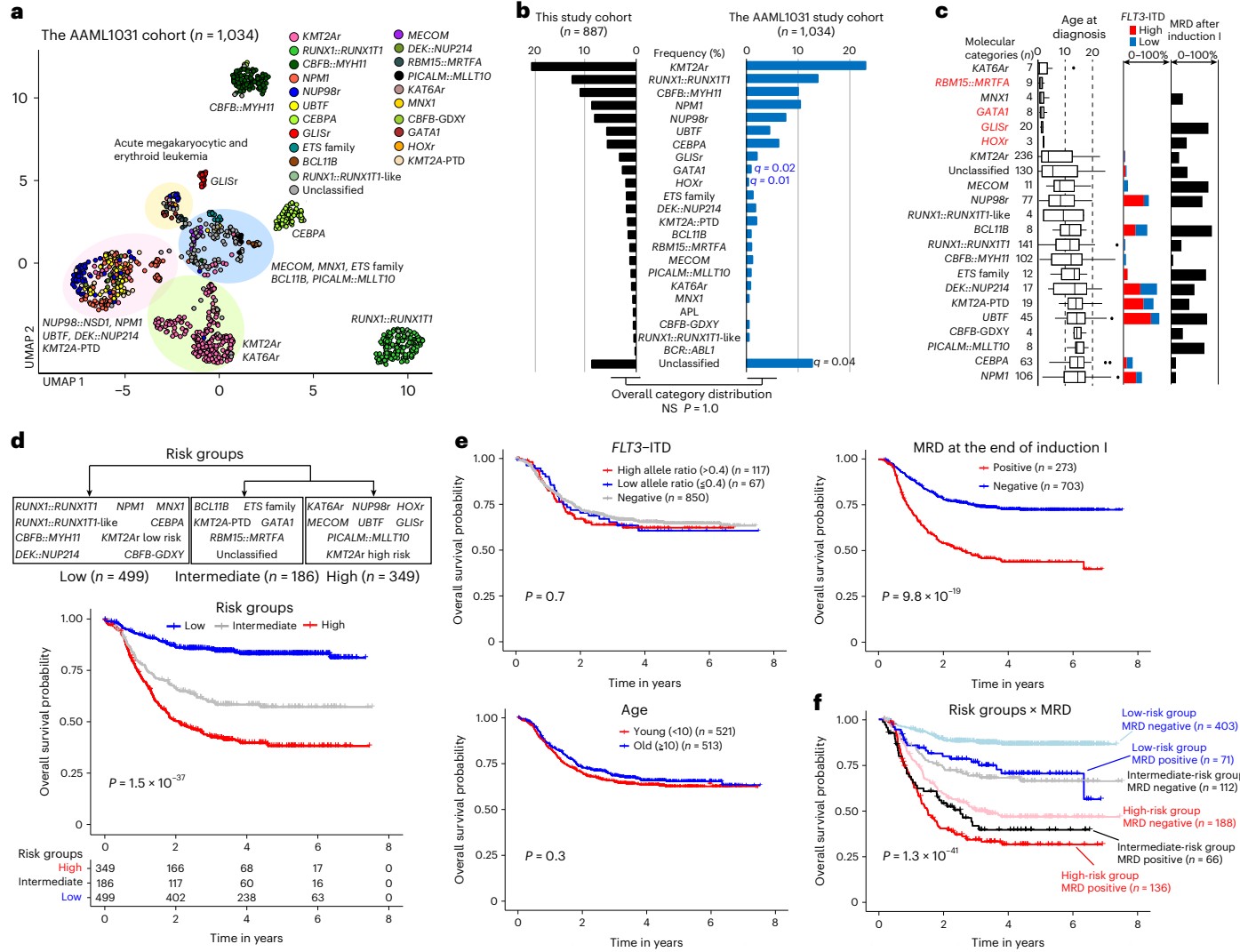

**Fig. 7 | Clinical association of molecular categories. a**, UMAP plot of transcriptome data of the AAML1031 cohort (*n* = 1,034) using top 340 variable genes. Dot colors denote molecular categories assigned to the samples according to genomic profiling using the same pipeline as this study cohort. Representative category names are shown, and large clusters enriching specific categories are highlighted in circles (pink: *NUP98::NSD1, NPM1, UBTF, DEK::NUP214, KMT2A-PTD*; green: *KMT2A*r and *KAT6A*r; yellow: categories with acute megakaryocytic or erythrocytic expression; blue: *MECOM, MNX1, ETS* family, *PICALM::MLLT10, BCL11B*). **b**, Frequency of molecular categories in the study cohort (black) and AAML1031 cohort (blue). The statistical significance of the frequency of each category assessed by two-sided Fisher's exact test followed by Benjamini–Hochberg adjustment (*q* < 0.05; blue indicates fewer and black indicates more in the AAML1031) is shown. **c**, Clinical features of molecular categories showing age at diagnosis (left), *FLT3*-ITD status (middle) and MRD positivity at the end of induction (right). Molecular category names associated with megakaryocytic

phenotypes are highlighted in red. The lines of the boxes represent the 25% quantile, median and 75% quantile. The upper whisker represents the higher value of maxima or 1.5× i.q.r., and the lower whisker represents the lower value of minima or 1.5× i.q.r. **d**, Grouping of molecular categories into low, intermediate and high-risk groups by recursive partitioning (upper) and Kaplan–Meier curves of overall survival of patients in each risk group (lower). **e**, Kaplan–Meier curves and statistical significance of overall survival of patients with known prognostic factors (*FLT3*-ITD status (upper left), age (lower left) and MRD positivity at the end of the induction I (upper right)). **f**, Kaplan–Meier curves of overall survival of patients in six risk strata using risk groups (low–intermediate–high) and MRD positivity. For survival curves in **d**, **e** and **f**, statistical significance was assessed by the log-rank test, and *P* values are shown in the plot. For survival analysis involving MRD status, patients with available MRD status (MRD+: *n* = 273; MRD−: *n* = 703) are included. NS, not significant.

(Fig. 6e). Given that germline mutations of *RUNX1* or *ETV6* are associated with leukemia with incomplete penetrance[60,61], these data suggest somatic alterations of these genes also require additional mutations for leukemia development, which may cooperatively define the immature leukemic phenotypes. Further accumulation of genomic data and experimental models will be necessary to understand immature pAML with these mutations.

**Clinical association of molecular categories**

Although the association between *KMT2A*r or *NUP98*r and poor outcomes is well-appreciated, the clinical associations of new molecular

categories have been discussed only in separate studies[8,25]. To address this deficiency and translate them into a clinical framework, we investigated the outcomes of these molecular categories using the COG AAML1031 study[13] (*n* = 1,034; Supplementary Table 22). Analyses of the AAML1031 RNA-seq data using the same pipeline revealed similar clustering of molecular categories and the overall category frequencies (Fig. 7a,b). The AAML1031 cohort confirmed the association of molecular categories with age and *FLT3*-ITD status, and showed variable minimal residual disease (MRD) positivity among molecular categories (Fig. 7c). Major categories with favorable outcomes aligned with previous reports (for example, *RUNX1::RUNX1T1* (*n* = 141), *CBFB::MYH11*

($n$ = 102) and *CEBPA* ($n$ = 63); Extended Data Fig. 9a). We also confirmed the known association of *GLIS*r[7] ($n$ = 20), *MECOM* ($n$ = 11), *PICALM::MLLT10* ($n$ = 8) and *KAT6A*r ($n$ = 7) with poor outcomes, except *DEK::NUP214* ($n$ = 17) which showed a favorable outcome in the AAML1031 study[29,62]. New categories of *MNX1* ($n$ = 4), *RUNX1::RUNX1T1*-like ($n$ = 4) and *CBFB*-GDXY ($n$ = 4) showed favorable outcomes.

We also investigated the clinical association of molecular heterogeneities within major categories. Among *KMT2A*r, fusion partners or *MECOM* expression[63,64] also confounded in the AAML1031 cohort (Extended Data Fig. 9b). Cox hazard models showed that both fusion partners and expression clusters are prognostic ($P$ = 0.00052 and 0.0015, respectively), with fusions with *SEPTIN* family and *MLLT11* or immature expression patterns associated with favorable outcomes (Extended Data Fig. 9c). The association of fusion partners or expression clusters with prognosis did not significantly differ (difference in C-index of 95% bootstrap interval for fusions and clusters: −0.025 to 0.093). Although *HOXB* categories of *NUP98*r, *NPM1* and *UBTF* also showed heterogeneity of expression patterns, their outcomes were not associated with UMAP clusters or fusion partners (Supplementary Fig. 2a).

Given these findings, we next applied recursive partitioning models[65] for censored event time data of molecular categories and fusion partners of *KMT2A*r, which revealed three groups with distinctive prognoses (Fig. 7d and Supplementary Fig. 2b–d). Univariate analyses revealed that age and *FLT3*-ITD were not prognostic, which could reflect the sorafenib given to patients with high-allelic *FLT3*-ITD in the AAML1031 study[13] (Fig. 7e). Contrarily, MRD positivity and a subset of cellular hierarchy scores were associated with overall survival (Fig. 7e and Extended Data Fig. 9d). A Cox proportional hazards model using risk groups and prognostic factors showed that hierarchy scores did not significantly contribute to prognosis, whereas risk groups and MRD positivity were independently prognostic (Supplementary Table 23). These data led us to establish a simple predictive framework solely based on molecular categories and MRD positivity, resulting in six risk strata with granular outcome prediction (Fig. 7f and Extended Data Fig. 9e). The prognostic values were validated using the separate AML08 trial[12] cohort ($n$ = 221; Extended Data Fig. 10a–c and Supplementary Tables 24 and 25). Hematopoietic stem cell transplantation in the first remission showed a benefit for high-risk categories with MRD, whereas that for the remaining groups needs further assessment (Extended Data Fig. 10d). Also, the predictive value of this prognostic framework was comparable or superior to various risk stratifications currently used in clinical trials for pAML[13–15] or ELN2022 (ref. 11) for adult AML (Supplementary Fig. 3). These data suggest that the proposed framework could be a basis for future risk stratification and clinical decisions.

## Discussion

In addition to known enrichment of chromosomal events like t(11,x) in pAML, sequencing technologies have identified additional pediatric-enriched driver alterations[7,8,27]. This prompted us to comprehensively investigate the increasingly complex genomic landscape of pAML in the context of the latest classification systems for hematological malignancies (WHO[5th] (ref. 9) and ICC[10]) and to develop a pAML-focused categorization. In this study, we systematically categorized our pAML cohort of 887 patients using an approach based on RNA-seq, resulting in 23 molecular categories defined by mutually exclusive driver alterations, covering 91.4% of the entire cohort. Of these 23 categories, 12 are not currently defined by WHO[5th]. These include common categories like *UBTF*, *GLIS*r and *GATA1*, otherwise categorized as 'AML-MR' or 'acute myeloid leukemia with other defined gene alterations' in the current WHO classification. Notably, myelodysplasia-related mutations or chromosomal alterations often co-occur with many pAML category-defining alterations and override them in WHO[5th] or do not drive consistent gene expression patterns

even without category-defining alterations. Considering that the current classification systems are mainly based on evidence from adult AML, we propose an alternative framework for pAML to better reflect its biology.

These molecular categories show unique expression and mutational profiles, whereas some categories also show critical similarities, which can suggest common molecular mechanisms and potential therapeutics. In particular, we noticed two large clusters characterized by *HOXA-B* expression profiles. Molecular categories with *HOXB* signatures were strongly associated with *FLT3*-ITD and *WT1* mutations, whereas those with *HOXA* signatures were associated with *KRAS* mutations. Considering that AMLs with *KMT2A*r, *NUP98*r and *NPM1* are dependent on KMT2A/Menin[66–68] and that several Menin inhibitors targeting *KMT2A*r and *NPM1* AML are in clinical trials[69,70], our data suggest that other subtypes marked by HOX expression may also be candidates for Menin inhibitors. This is supported by our recent study showing that *UBTF* AMLs are sensitive to Menin inhibitors[71]. Also, the high frequency of *FLT3*-ITD in categories with *HOXB* expression implies that FLT3 signaling is closely related to biology and that treatment with FLT3 inhibitors for *FLT3*-ITD[+] *HOXB* subtypes independent of the allelic ratio may be effective.

Some cases without category-defining alterations could be characterized by rare fusion or mutations, which need further evidence to establish as a disease entity, including *MLLT1* and *H3F3A* mutations that are frequent and class-defining in Wilms tumor[57] and glioma[58], respectively. Considering that AML and Ewing sarcoma also share *ETS* family fusions[40] (for example, *EWSR1::ERG*), it would be intriguing to incorporate knowledge of these solid tumors to understand the biology behind pAML with these rare alterations. Also, enrichment of *RUNX1* or *ETV6* loss-of-function alterations in immature AML implies that these can be class-defining in the absence of other defining alterations and likely with specific cooperating mutations. These findings further suggest a continuum with other immature leukemias, such as early T cell precursor-ALL and mixed phenotype acute leukemias (T/My) with similar mutational features[72,73].

We further investigated the clinical outcomes of these molecular categories using two independent cohorts: the COG AAML1031 study and the AML08 study. Using both cohorts, we show a strong association of new molecular categories with outcomes (for example, *PICALM::MLLT10, UBTF* and *KAT6A*r as high risk, and *CBFB*-GDXY as low risk). These analyses also revealed that molecular categories and known prognostic factors, such as *FLT3*-ITD status or cellular hierarchy scores, are confounding. With this comprehensive profiling recognizing new pAML subtypes, we established a simple risk stratification using molecular categories and MRD. This strategy, however, heavily relies on the analysis of next-generation sequencing data. Although the WHO classification requires targeted sequencing or WGS, we propose a diagnostic pipeline utilizing RNA-seq, which is highly sensitive for canonical and cryptic fusion calling, allows for categorization based on gene expression signatures, including outlier and allele-specific expression (*MECOM*, *BCL11B* and *MNX1*), and provides limited but sufficiently sensitive mutation calling to enable our comprehensive molecular categorization strategy to newly diagnosed pAML. This approach is favored over current commercial panels commonly used for pAML, which either lack coverage of all the defining genes (for example, *UBTF*) or are unsuitable for detecting complex structural variations that drive aberrant expression of *MECOM* or *BCL11B*. Given that clinical sequencing is not readily available globally and these molecular analyses require substantial expertise, robust and easy pipelines are needed for future and broad application of this framework for pAML in the general clinical setting.

## Online content

Any methods, additional references, Nature Portfolio reporting summaries, source data, extended data, supplementary information,

acknowledgements, peer review information; details of author contributions and competing interests; and statements of data and code availability are available at https://doi.org/10.1038/s41588-023-01640-3.

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

## Methods

### Subject cohorts and sample details

Tumor samples from patients with AML from the St. Jude Children's Research Hospital tissue biorepository were obtained with written informed consent from patient, parents or guardians using a protocol approved by the St. Jude Children's Research Hospital institutional review board. Studies were conducted in accordance with the International Ethical Guidelines for Biomedical Research Involving Human Subjects. No patient received compensation for the enrollment to this study. Samples for RNA-seq ($n$ = 221), WGS ($n$ = 58) and WES ($n$ = 7) are newly sequenced in this study, and the rest of the data were obtained from previous publications[4,7,8,16–25] or public databases (see details in 'Data availability' and Supplementary Table 1). For samples with multiple available data points, we included one representative time point with a high tumor purity and good RNA-seq data quality. Cases were assigned to current WHO[5th] (ref. [9]) and ICC[10] by board-certified hematopathologists (P.K. and J.M.K.).

### Genotype fingerprints

To make sure that the study cohort cases represent unique individuals, we performed a pairwise genotype concordance comparison among all the study cases using the estimated genotype from single nucleotide polymorphisms (SNPs) with ≥20 coverage in RNA-seq Binary Alignment Map (BAM) files. We set genotype concordance percentage cutoff at ≥90% of SNPs shared between two individuals to identify potential duplicates, confirming the uniqueness of the 887 patients in the study cohort.

### Sample processing, library preparation and sequencing

For newly sequenced samples with low tumor purity (<60%), the leukemic cell population was enriched either by flow cytometric sorting or T cell depletion by magnetic beads (EasySep Human CD3 Positive Selection Kit II; StemCell Technologies, catalog no. 17851). For flow cytometric sorting, CD45[dim]CD33[dim] positive population was sorted using anti-CD45 PerCP-Cyanine5.5 (eBioscience, catalog no. 8045-9459-120; 1:20 dilution), anti-CD33 APC (eBioscience, catalog no. 17-0338-42; 1:20 dilution) and DAPI (BD Biosciences, catalog no. 564907) using FACS Aria III instrument and FACS Diva v.9.0 (both BD Biosciences) (Extended Data Fig. 1c). CD34 gating using anti-CD34 PE (phycoerythrin) (Beckman, catalog no. IM1459U; 1:5 dilution) was added depending on the positivity of each patient sample. Enrichment of the tumor population was confirmed by flow cytometric analysis of the postsorting samples (generally >90%). Libraries were constructed using the TruSeq Stranded Total RNA Kit, with Ribozero Gold (Illumina, catalog no. 20020598) for RNA-seq, the TruSeq DNA PCR-Free Library Prep Kit (Illumina, catalog no. 20015963) for WGS and the TruSeq Exome Kit v.1 (Illumina, catalog no. 20020614) for WES according to the manufacturer's instructions. After library quality and quantity assessment, samples were sequenced on HiSeq2000 or 2500 (Illumina, RRID:SCR_020132, RRID:SCR_016383) instruments with paired-end (2 × 101 bp, 2 × 126 bp or 2 × 151 bp) sequencing using TruSeq SBS Kit v3-HS (Illumina, catalog no. FC-401-3001) or TruSeq Rapid SBS Kit (Illumina, catalog no. FC-402-4023) and HiSeq Control Software with most recent version at the time of sequencing.

### RNA-seq mapping, fusion detection and large-scale CNV calling

RNA reads from newly sequenced samples and from publications were mapped to the GENCODE (RRID:SCR_014966) human genome assembly release 19 gene annotation (GRCh37/hg19) using the StrongARM pipeline[74]. Chimeric fusion detection was carried out using CICERO[75] (v.0.3.0). For the cases with only RNA-seq data, RNAseqCNV[76] (v.1.2.1) was used to call large-scale CNV.

### Somatic mutation calling from RNA-seq

To detect SNV and indel from RNA-seq data, we applied the following approach to simultaneously account for germline polymorphisms (without germline control) and sequencing artifacts specific to RNA-seq on a panel of 87 predefined genes previously reported to be significantly mutated in pAML[4] and myelodysplastic syndrome (Supplementary Table 5). Briefly, candidate SNVs/indels were called by Bambino[77] (v.1.07) or RNAindel[78,79] (v.3.0.4), annotated by VEP[80] (v.95), filtered by excluding variants with gnomAD (v.2.1.1, RRID:SCR_014964)[81] population allele frequency >0.1% as possible germline variants, and in turn, classified for putative pathogenicity with PeCanPie/MedalCeremony[82] (not versioned). Candidate variants with putative pathogenicity were considered germline or artifacts if present in >5% of the cases. Candidate variants were further filtered if the number of supporting reads was ≤5 or if the VAF was ≤5%. *UBTF* tandem duplications were detected by CICERO focusing ITD or PTD with supporting reads ≥3 within exon 13 of *UBTF* gene or adjacent introns and CICERO score <10, detection of indels on exon 13 of the *UBTF* gene, and counting reads with 10 or more soft-clipped nucleotide sequences and total reads on the 3′-end of exon 13 that contains a hotspot of ITD and PTD (GRCh37-lite, chr17:42288162-42288192; GRCh38, chr17: 44210794-44210824)[8].

### WGS and WES data analysis

The previous genomic lesion calls for the cases (WGS: $n$ = 394; WES: $n$ = 284) from published studies[4,7,8,16,18–20,23,25] were collected from their respective publications. For the unpublished cases with DNA data (WGS: $n$ = 136; WES: $n$ = 107), DNA reads were mapped using BWA[83,84] (WGS: v0.7.15-r1140 and v0.5.9-r26-dev; WES: v0.5.9-r26-dev and v0.5.9, RRID:SCR_010910) to the GRCh37/hg19 human genome assembly. Aligned files were merged, sorted and de-duplicated using Picard tools 1.65 (broadinstitute.github.io/picard/). SNVs and indels were called using Bambino. For cases paired with matched germline controls, germline variants were filtered out if present in the matched germline sample. For unpaired cases, possible germline variants were filtered and classified as for somatic mutation calling from RNA-seq. The counting of somatic mutations included all the pathogenic or likely pathogenic mutations detected by WGS, whereas mutation detection from cases with only RNA-seq data is limited to the 87 preselected genes. SVs were analyzed using CREST (Clipping REveals STructure)[85] (v.1.0), and CNVs were analyzed using CONSERTING[86] on the WGS data. CNVs were also called on cases with only WES DNA data using the following methods. Briefly, Samtools[87] (v.1.16) mpileup command was used to generate a mpileup file from matched germline and tumor BAM files with duplicates removed. If a matched germline was not available, a high-quality normal sample was used to pair with the tumor sample. VarScan[88] (v.2.3.5) was then used to take the mpileup file to call somatic CNVs after adjusting for normal/tumor sample read coverage depth and GC content. Circular Binary Segmentation algorithm[89] implemented in the DNAcopy R package (v.1.52.0) was used to identify the candidate CNVs for each sample. B-allele frequency information was also used to assess allelic imbalance.

### Validation of somatic alterations called by the RNA-seq pipeline

We focused on 243 cases (27.4%) with data from all three platforms (matched WGS, WES and RNA-seq) to cross-validate the accuracy of our RNA-seq based pipeline[8]. Of 374 SNV/indel variant calls from RNA-seq data, 329 variants (88%) were called from either WGS or WES, whose VAFs showed significant correlation with those of RNA-seq calls (Extended Data Fig. 1f). Of the remaining 45 calls, 35 have supporting reads in DNA data, which were not called, likely because of sequence noises and low VAF, validating in total 97.3% of the RNA-seq calls.

### GRIN analysis for significantly mutated genes

For the 887 AML cases, the GRIN (v.2.0) model[26] was used to evaluate the statistical significance of the number of subjects with each type of

lesion: fusions, CNVs (amplifications and deletions), copy-neutral loss of heterozygosity, SNV/indels and tandem duplications in each gene. For each type of lesion, robust false discovery estimates were computed from $P$ values using Storey's $q$ value[90] with the Pounds–Cheng estimator of the proportion of hypothesis tests with a true null hypothesis[91]. A false discovery rate (FDR) cutoff of <0.05 was used to obtain significantly mutated genes, where we focused on protein-coding genes and genes that are known or likely to be pathogenic in leukemia. We also excluded genes that are part of a large chromosomal gain, loss or copy-neutral loss of heterozygosity but not the target of the CNVs based on Genomic Identification of Significant Targets in Cancer (GISTIC) analysis. Subgroup GRIN analyses for *HOXA* categories ($n = 164$), *HOXB* categories ($n = 207$) categories and the Unclassified category ($n = 76$) were similarly performed.

## GISTIC analysis for significant recurring copy-number alterations

We used GISTIC (v.2.0.23, RRID:SCR_000151)[92,93] to identify genomic regions that are significantly amplified or deleted across our 895 samples. Each aberration was assigned a G-score that considered the amplitude of the aberration as well as the frequency of its occurrence across samples. FDR $q$ values were then calculated for the aberrant regions, and regions with $q$ values ≤0.25 were considered significant. A 'peak region' was identified for each significant region with the greatest amplitude and frequency of alteration. In addition, a 'wide peak' was determined using a leave-one-out algorithm to allow for errors in the boundaries in a single sample. Each significantly aberrant region was also tested to determine whether it resulted primarily from broad or focal events (a broad event was set as >90% of the chromosome arm, whereas a focal event was ≤90%).

## Allele-specific expression estimation for *MNX1*, *BCL11B* and *MECOM* categories

For cases with both WGS and RNA-seq available, SNP markers in the respective gene locus with ≥10x coverage that are heterozygous (defined as $0.2 ≤ VAF ≤ 0.8$) in WGS and present in RNA-seq were extracted, and a two-sided binomial test (with probability of success $P = 0.5$) was performed on each marker for allelic imbalance in RNA expression. The median of binomial $P$ values was used to assess allele-specific expression. For RNA-seq only cases, SNP markers in the respective gene locus with ≥10x coverage and allelic imbalance ($VAF ≤ 0.2$ or $VAF ≥ 0.8$) support allele-specific expression.

## Germline variant curation methods

We focused on 15 candidate genes relevant to AML that define specific categories in WHO[5th] (Supplementary Table 26) and scanned for germline mutations in the cases with WGS or WES germline BAM files available (WGS: $n = 367$; WES: $n = 354$). For cases with germline mutation called in previously published studies[8,21], we collected calls from the studies. For the remaining cases, the putative germline variants were called using Bambino, annotated by VEP, and classified for putative pathogenicity with PeCanPie/MedalCeremony. We then used the following criteria to obtain the candidate germline variants: gnomAD population allele frequency ≤0.001; read coverage SNV ≥ 20 and indel ≥ 15; for SNV, VAF between 0.2 and 0.8; for indel, ≥3 reads supporting the alternative allele. All candidate germline variants were comprehensively reviewed and classified based on recommendations from the American College of Medical Genetics and Genomics and the Association for Molecular Pathology[94] and the Clinical Genome Resource[95–98] by a variant scientist (J.L.M.).

## Inference of genetic ancestry

For each individual, the admixture fraction was estimated using the iAdmix program[99] and allele frequencies from the 1000 Genomes Project reference populations (European (EUR), African (AFR), Native American (NA), East Asian (EAS), South Asian (SAS)) were used as a reference[100]. Overall, the genetic ancestral composition for each single individual was derived based on a comparison of allele frequencies between each individual and reference genome. The sum of coefficients from the five populations was assumed to sum to 100%. An RNA-seq BAM file was used as input directly to iAdmix program, where allele frequencies for the coding SNPs from the 656,129 SNPs were used in the ancestry estimation. The categorization of individuals into ancestral groups was performed based on the composition of genetic ancestry estimated from iAdmix program (Black: AFR > 70%; East Asian: EAS > 90%; Hispanic: NA > 10% and NA greater than AFR; South Asian: SAS > 70%; White: EUR > 90%). The remaining patients with majority EAS or SAS were categorized into 'Other-Asian', and the rest of patients with majority EUR or AFR or NA > 10% with NA less than AFR, were categorized into 'Other-US'[101] (Supplementary Table 1).

## Gene expression data summarization, batch correction, dimension reduction and clustering

Reads from aligned BAM files were assigned to genes and counted using HTSeq[102] (v.0.11.2, RRID:SCR_005514) with the GRCh37/hg19 GTF file. For a gene to be considered as expressed, we required that at least five samples should have ≥10 read counts per million (cpm) reads sequenced. The count data were transformed to $\log_2(cpm)$ using Voom[103] available from R package Limma[104] (v.3.50.3, RRID:SCR_010943). We corrected for library strand (stranded total RNA versus unstranded messenger RNA) and batch effect between the TARGET and the rest of cohorts using the ComBat method available from R package SVA[105] (v.3.42.0, RRID:SCR_012836). The R package Seurat[106–109] (v.4.1.0, RRID:SCR_016341) was used for dimension reduction and sample clustering. Briefly, the top 315 variable genes were selected using the 'vst' method. The expression data were then scaled and used for principal component analysis, and the top 100 principal components were used for dimension reduction using UMAP[110,111] (RRID:SCR_018217) (n_neighbors = 12 and min_dist = 0.2). Samples were clustered using the top 100 principal components by first constructing a K nearest-neighbor graph and then iteratively optimizing the modularity using Louvain algorithm (resolution = 3.5). Dimension reduction was also performed by Diffusion maps[37,112] algorithm available in the R package destiny[113] (v.3.10.0) using the same 315 genes with the default setting except for number of principal components (n_pcs = 50).

Differential gene expression analysis was performed by Limma[104], and we set $\log_2(cpm) = 0$ if it is <0 based on the $\log_2(cpm)$ data distribution. $P$ values were adjusted by the Benjamini–Hochberg method to calculate the FDR using the R function p.adjust. Genes with absolute fold change >2 and FDR <0.05 were regarded as significantly differentially expressed. GSEA[114] was performed by GSEA v.4.2.3 (RRID:SCR_003199) using MSigDB gene sets c2.all (v.7.5.1), comparing each category with the rest of the categories. Permutations were done 1,000 times among gene sets with sizes between 15 and 1,500 genes. Normalized enrichment scores and FDR for arbitrary gene sets representing hematopoiesis, leukemia phenotype, biological processes and drug responses were shown. WGCNA was carried out by R package WGCNA[47] (v.1.70-3, RRID:SCR_003302) using the top 2,000 variable genes and default setting with the exception of block-wide module calculation with reassignThreshold = 0 and mergeCutHeight = 0.25. Functional annotations of the top 315 variable genes, differentially expressed genes and genes in WGCNA modules were performed with DAVID[115] (v.6.8), and results for the Gene Ontology term, biological process (GOTERM_BP_DIRECT) were exported. Inference of cellular hierarchy by CIBERSORT[116] (RRID:SCR_016955) was performed by the web interface of CIBERSORTx in absolute mode with S-mode batch correction without a permutation[36]. Transcript per million values and Malignant Signature Matrix and Malignant Single Cell Reference Samples from a publication[36] were used as input files, and the malignant cell populations were normalized

to 1 to calculate the relative fraction scores, which were shown in UMAP space or violin plots. Prognostic scores of LSC17[48], pLSC6 (ref. 49), ADE-RS[117] and iScore[46] were calculated as reported. Hierarchical clustering (RRID:SCR_014673) of expression data, mutual-exclusivity matrix and GSEA scores was performed using the Euclidian distance and Ward method with pheatmap (v.1.0.12, RRID:SCR_016418).

## Statistics and reproducibility

No sample size, power calculation or randomization of patients was performed in this study utilizing retrospective profiling of patients with available materials or sequence data. No analysis depending on patient background was performed in this study. No blinding was performed in the enrollment of patients or data collection of public data, and blinding in group allocation was not possible because the grouping is based on the molecular characteristics of individual patients. For discrete values of the molecular category and the mutation frequency in cohorts, statistical significance and mutual exclusivity were assessed by two-sided Fisher's exact test and Pearson's correlation. Adjustment of multiple testing was performed by the Benjamini–Hochberg method using the p.adjust function in R when appropriate. For survival data, decision trees were established by a recursive partitioning method using R library rpart[65] (v.4.1.19, RRID:SCR_021777). Kaplan–Meier curves for the probability of overall survival and event-free survival were constructed using the R package survival (v.3.3-1, RRID:SCR_021137). Events in the probability of event-free survival calculations were defined as relapse, death in remission by any cause and nonresponse, which was included as an event at the date of diagnosis. The Cox proportional hazards model was used to calculate the hazard ratio. The log-rank test (two-sided) was used to calculate the statistical significance of individual prognostic factors by univariate analyses first, and significant factors were included in a multivariate analysis. Clinical association of the molecular categories was first assessed using the AAML1031 study (NCT01371981, $n$ = 1,034), and the results were validated using the AML08 cohort (NCT00703820, $n$ = 221, independent from the AAML1031, a part of this study cohort). We quantified the predictiveness of recursive partitioning survival tree models and risk classification systems with Harrel's concordance index for Cox models[118] using a bootstrap procedure. We generated 1,000 bootstrap datasets by sampling patients with replacement and computed concordance index values for each bootstrap dataset. The 2.5 and 97.5 percentiles were used to define the bootstrap confidence interval endpoints. Concordance index values of a pair of risk classification systems were similarly computed similarly. Regression tree models were refit to each bootstrap dataset in the model development analysis on the AAML1031 cohort. For all other analyses, the risk classification was defined externally from the cohort and thus risk-group definitions for individual patients remained constant across bootstrap datasets. R statistical environment (R v.4.0.2, RRID:SCR_001905) was used for statistical tests.

**Visualization.** Mutational heatmaps and mutations on individual genes were visualized using ProteinPaint (proteinpaint.stjude.org/). Heatmaps of expression data, mutual-exclusivity matrix and GSEA scores were created by pheatmap function. Other data visualizations were performed by ggplot function of R library ggplot2 (v.3.3.6, RRID:SCR_014601), survminer (v.0.4.9) and base plot function in R statistical environment. Figures are incorporated and edited using Adobe Illustrator (2021, RRID:SCR_010279). Annotation of genes in mutational heatmaps depends on common knowledge, and the definition of RAS pathway genes included causative genes of Noonan or Noonan-like syndrome[119].

## Reporting summary

Further information on research design is available in the Nature Portfolio Reporting Summary linked to this article.

## Data availability

Genomic analyses in this study are based on the GENCODE GRCh37/hg19, and gnomAD v.2.1.1 was used for classification for germline and somatic mutations. The genomic data and expression data newly generated in this study (RNA-seq: $n$ = 221, WGS: $n$ = 58, WES: $n$ = 7) have been deposited in the European Genome-Phenome Archive (EGA, RRID:SCR_004944), which is hosted by the European Bioinformatics Institute (EBI), under accession EGAS00001005760. Subsets of the new data (RNA-seq: $n$ = 221, WGS: $n$ = 53, WES: $n$ = 5) have been also deposited to St. Jude Cloud under Pan-AML study (https://permalinks.stjude.cloud/panaml). Details are found in Supplementary Table 1. For previously published RNA-seq data ($n$ = 393), 266 are available either on EGA or St. Jude Cloud[7,8,17,19–23,25] or from the original publication[24]. For the other 127 published cases[18], we downloaded the BAM files from EGA (EGAS00001004701). For previously published WGS data ($n$ = 198), 106 from the original publications[7,8,19,20,23,25] are available on either EGA or St. Jude Cloud, and the other 92 published BAM files[18] were downloaded from EGA (EGAS00001004701). For the previously published WES data ($n$ = 273), 153 with data from the original publications[7,8,17,19–23,25] are available either on St. Jude Cloud or EGA, and the BAM files for the other 120 published cases[18] were downloaded from EGA (EGAS00001004701). We also downloaded data for publicly available but previously unpublished RNA-seq data ($n$ = 86) on St. Jude Cloud under the PCGP study (https://permalinks.stjude.cloud/permalinks/PCGP, $n$ = 8) and the RTCG study (https://platform.stjude.cloud/data/cohorts?dataset_accession=SJC-DS-1007, $n$ = 78). Similarly, we obtained unpublished WGS data ($n$ = 82: RTCG) and WES data ($n$ = 2: PCGP, $n$ = 99: RTCG study). The data generated by the TARGET initiative[4,16] ($n$ = 187), including additional samples from the AAML1031 trial[13] ($n$ = 1,034), are also available under accession phs000218 (TARGET-AML) and phs000465 (TARGET substudy, data is available as a part of phs000218), managed by the NCI, and were obtained through GDC Portal managed by NCI under the TARGET-AML study (https://portal.gdc.cancer.gov/projects/TARGET-AML). Information about TARGET can be found at http://ocg.cancer.gov/programs/target. These sequencing data are available through controlled access as part of the NIH Genomic Data Sharing Policy (https://grants.nih.gov/grants/guide/notice-files/NOT-OD-14-124.html) and data access is restricted for academic use. Source data are provided with this paper.

## Code availability

We did not use custom code or software for this study.

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

## Acknowledgements

We thank all the patients and their families at St. Jude Children's Research Hospital (SJCRH) for their contribution to the biological specimens used in this study. We also thank the Biorepository, the Flow Cytometry and Cell Sorting Core, and the Hartwell Center for Bioinformatics and Biotechnology at SJCRH for their essential services. This work was funded by the American Lebanese and Syrian Associated Charities of SJCRH and grants from the National Institutes of Health (grant no. P30 CA021765, Cancer Center Support Grant and a Developmental Fund Award to J.M.K. and X.M., and grant no. U54CA243124, Fusion Oncoproteins in Childhood Cancers (FusOnC2) Consortium to J.M.K. (co-principal investigator). The content, however, does not necessarily represent the official views of the National Institutes of Health and is solely the responsibility of the authors. This work was also supported in part by the Fund for Innovation in Cancer Informatics (the-ici-fund.org, to X.M. and J.M.K.). J.M.K. holds a Career Award for Medical Scientists from the Burroughs Wellcome Fund and is a previous recipient of the V Foundation Scholar Award (Pediatric). The funders had no role in

study design, data collection and analysis, decision to publish or preparation of the manuscript.

## Author contributions

J.M.K., J.M., T.W. and M.U. conceptualized and managed the entire project. J.M.K., J.M., M.U., G.S. and M.P.W. performed mutational analyses. S.P., Y.N., T.A.A. and M.U. performed clinical outcome analyses. J.M.K. and J.L.M. reviewed and classified germline mutations. J.M.K. and P.K. reviewed mutational data and performed classification of the WHO and ICC. M.R., D.R., S.F., Y.L., W.Y., Y.F., G.W., X.M., B.J.H. and S.P. provided resources and software for data analysis. S.D.B., L.W., T.A.A. and J.E.R. provided data. M.U., J.M. and Y.N. prepared figures. M.U., J.M. and J.M.K. wrote the original draft of the manuscript. All authors reviewed and edited the manuscript. J.M.K. and S.P. supervised the project.

## Competing interests

The authors declare no competing interests.

## Additional information

**Extended data** is available for this paper at https://doi.org/10.1038/s41588-023-01640-3.

**Correspondence and requests for materials** should be addressed to Jeffery M. Klco.

**a** Cohorts

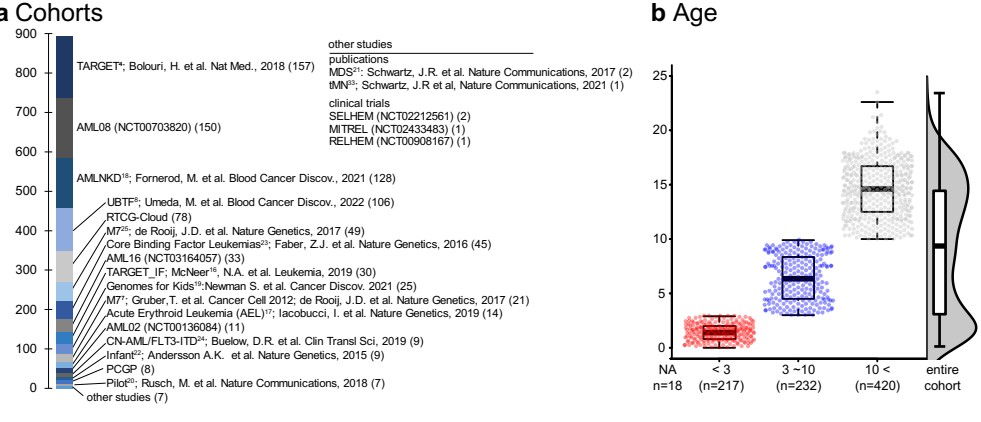

**b** Age

**c** Gating strategy for tumor population sorting

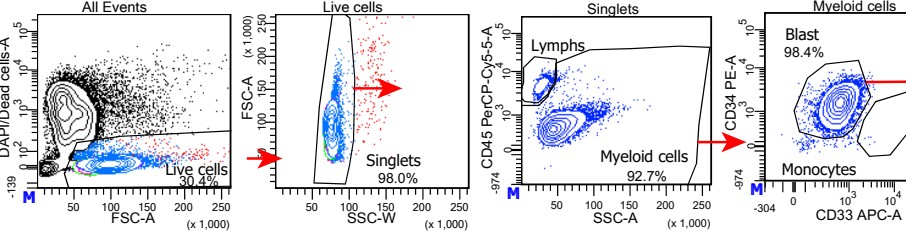

**d** Data availability

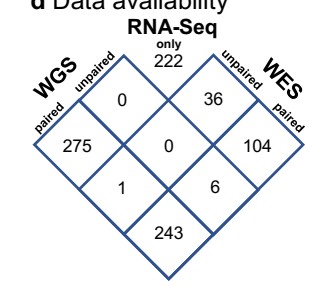

**e** Arm-level CNVs

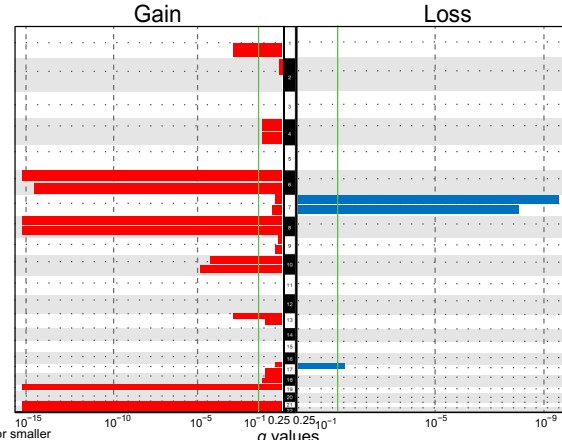

**g** Comparison with previous studies

| WHO classification (5th) | karyotype | AML-BFM 98[28] (N=454) | AML10-12[29] (N=633) | This study cohort (N=887) |
|---|---|---|---|---|
| | normal | 22.0% | 24.6% | 21.9% |
| Acute myeloid leukemia with *KMT2A* rearrangement | 11q23 | 20.0% | 16.4% | 20.2% |
| Acute myeloid leukemia with *RUNX1::RUNX1T1* fusion | t(8;21)(q22;q22) | 12.6% | 13.6% | 12.4% |
| Acute myeloid leukemia with *CBFB::MYH11* fusion | inv(16)(p13q22) | 9.3% | 6.8% | 10.8% |
| Acute myeloid leukemia with *DEK::NUP214* fusion | t(6;9)(p23;q34) | N.A | 1.6% | 1.8% |

reference 28: AML-BFM 98: J Clin Oncol. 2010 Jun 1;28(16):2682-9.
reference 29: AML10-12 : J Clin Oncol. 2010 Jun 1;28(16):2674-81.

**f** Cross-validation of mutation calls from the RNA pipeline

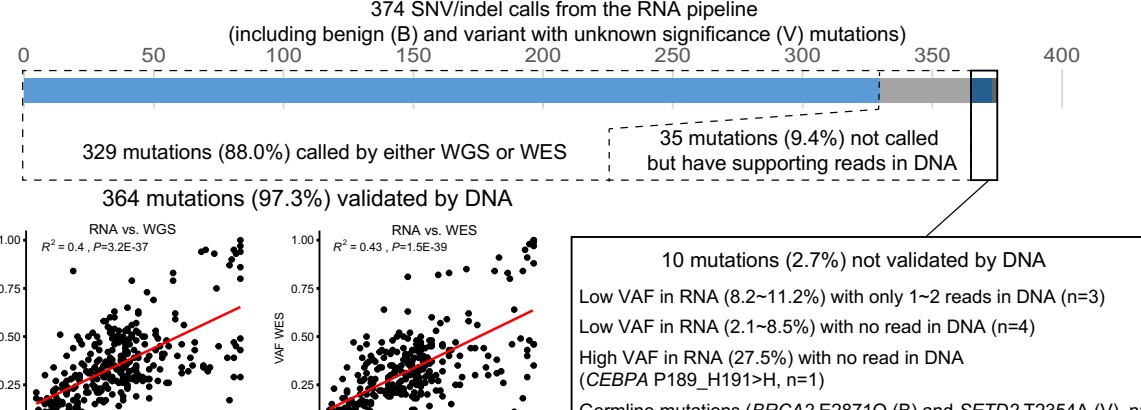

**Extended Data Fig. 1 | See next page for caption.**

**Extended Data Fig. 1 | Cohort details. a**. Data source of each patient with acute myeloid leukemia (AML), including publications and clinical trials. **b**. Age distribution of patients at diagnosis (red: age<3, blue: 3<age<10, gray: 10<age). Lines of the box represent 25% quantile, median, and 75% quantile. The upper whisker represents the higher value of maxima or 1.5 x interquartile range (IQR), and the lower whisker represents the lower value of minima or 1.5 x IQR. NA: not available. **c**. Representative gating strategy for sorting of the myeloid cell population. Vertical and horizontal axes are linear for FSC (forward scatter) and SSC (side scatter) and log-scaled for fluorescence-conjugated antibodies. CD34 gating was adjusted for individual patients depending on the positivity. **d**. A Venn diagram showing data platforms available for each patient. WGS: whole-genome sequencing, WES: whole-exome sequencing, RNA-Seq: RNA-sequencing. **e**. Results of GISTIC (Genomic Identification of Significant Targets in Cancer) analysis for arm-level chromosomal events. The left panel shows the enrichment of chromosomal gains (red), and the right panel shows the enrichment of chromosomal losses (blue). Green lines show a significance threshold for $q$ values (0.25). **f**. Cross-validation of single nucleotide variant (SNV) and insertion/deletion (indel) calls from the RNA pipeline using whole-genome/exome sequencing (WGS/WES) data. The bar graph shows mutation calls and the validation status. For those also called from DNA data, a comparison of variant allele frequency (VAF) and Pearson's correlation are shown in the bottom left, and the statistical test was performed as two-sided. A regression line is shown in red. For unvalidated calls, details are shown in the bottom right. **g**. A comparison of major classes of the World Health Organization (WHO) classification in the study cohort with karyotyping in previous large pediatric AML cohorts.

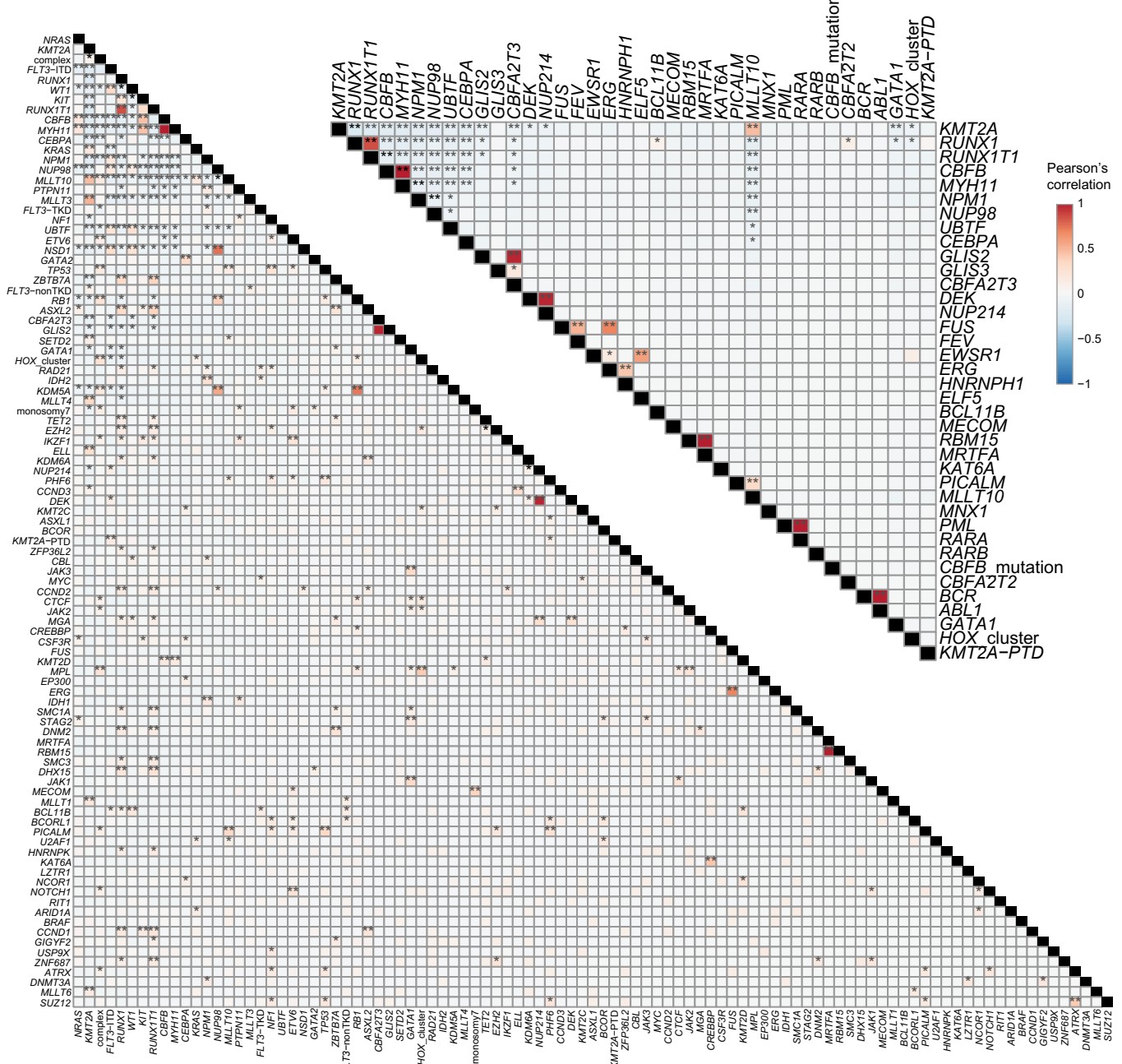

**Extended Data Fig. 2 | Mutational correlation.** Pair-wise correlation among most frequent 97 genetic alterations (n ≥ 5 in the entire cohort) from GRIN analysis and chromosomal changes (complex karyotype and monosomy 7) (**a**) and category-defining gene alterations (**b**). *KMT2A*-PTD (partial tandem duplication) is independently included from other *KMT2A* alterations, and *FLT3* alterations are classified into ITD (internal tandem duplication), TKD (tyrosine kinase domain) mutations, and non-TKD mutations due to the known functional difference. Colors correspond to Pearson correlation. Statistical significance was assessed by two-sided Fisher's exact test to calculate *P* values followed by the Benjamini-Hochberg adjustment for multiple testing to calculate *q* values (*$P < 0.05$, **$q < 0.05$).

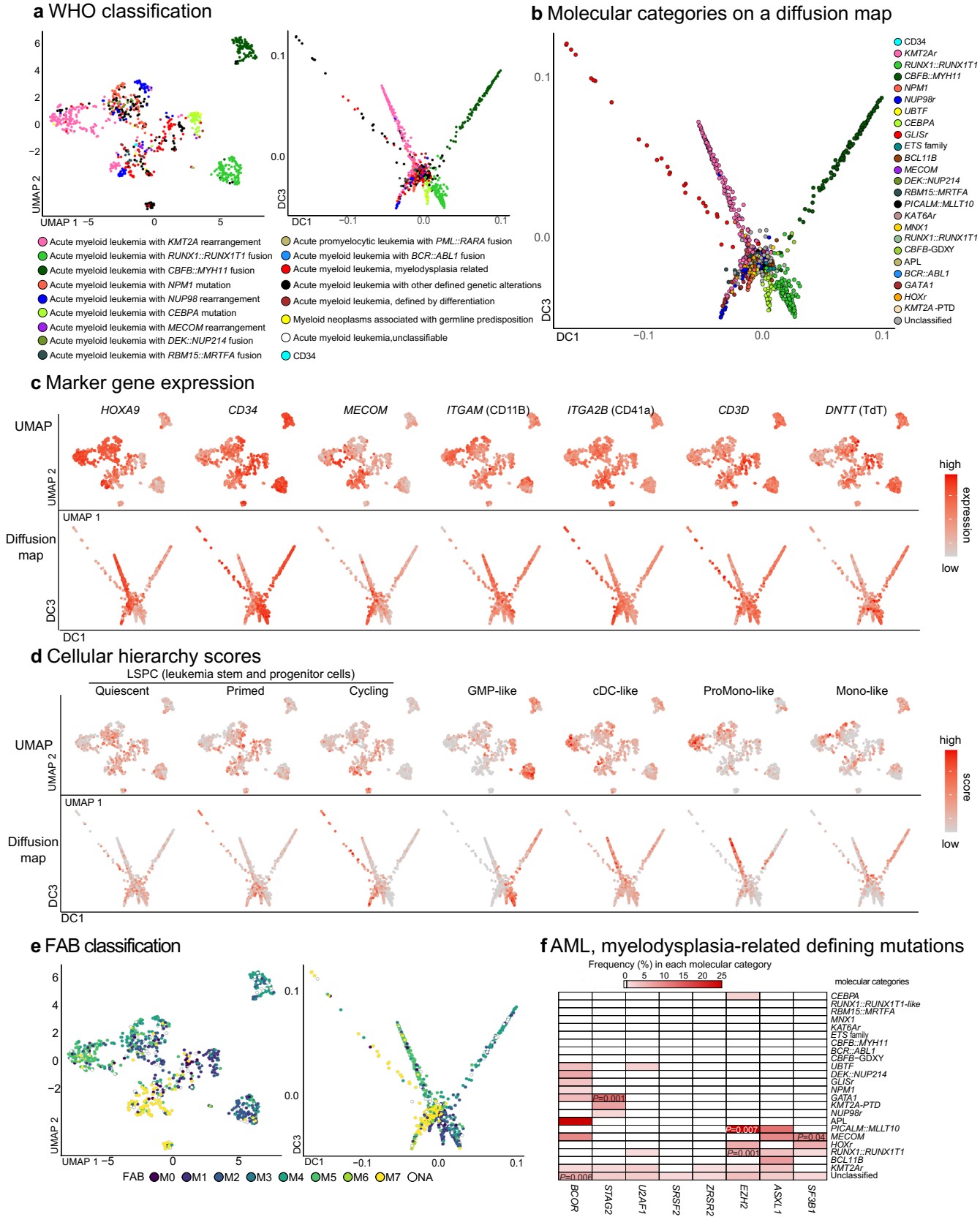

**a** WHO classification

Acute myeloid leukemia with *KMT2A* rearrangement
Acute myeloid leukemia with *RUNX1::RUNX1T1* fusion
Acute myeloid leukemia with *CBFB::MYH11* fusion
Acute myeloid leukemia with *NPM1* mutation
Acute myeloid leukemia with *NUP98* rearrangement
Acute myeloid leukemia with *CEBPA* mutation
Acute myeloid leukemia with *MECOM* rearrangement
Acute myeloid leukemia with *DEK::NUP214* fusion
Acute myeloid leukemia with *RBM15::MRTFA* fusion

Acute promyelocytic leukemia with *PML::RARA* fusion
Acute myeloid leukemia with *BCR::ABL1* fusion
Acute myeloid leukemia, myelodysplasia related
Acute myeloid leukemia with other defined genetic alterations
Acute myeloid leukemia, defined by differentiation
Myeloid neoplasms associated with germline predisposition
Acute myeloid leukemia,unclassifiable
CD34

**b** Molecular categories on a diffusion map

CD34
*KMT2Ar*
*RUNX1::RUNX1T1*
*CBFB::MYH11*
*NPM1*
*NUP98r*
*UBTF*
*CEBPA*
*GLISr*
*ETS* family
*BCL11B*
*MECOM*
*DEK::NUP214*
*RBM15::MRTFA*
*PICALM::MLLT10*
*KAT6Ar*
*MNX1*
*RUNX1::RUNX1T1*
*CBFB*-GDXY
APL
*BCR::ABL1*
*GATA1*
*HOXr*
*KMT2A* -PTD
Unclassified

**c** Marker gene expression

*HOXA9*  *CD34*  *MECOM*  *ITGAM* (CD11B)  *ITGA2B* (CD41a)  *CD3D*  *DNTT* (TdT)

UMAP

Diffusion map

**d** Cellular hierarchy scores

LSPC (leukemia stem and progenitor cells)
Quiescent  Primed  Cycling  GMP-like  cDC-like  ProMono-like  Mono-like

UMAP

Diffusion map

**e** FAB classification

FAB   M0  M1  M2  M3  M4  M5  M6  M7  NA

**f** AML, myelodysplasia-related defining mutations

Frequency (%) in each molecular category
0  5  10  15  20  25

molecular categories

*CEBPA*
*RUNX1::RUNX1T1-like*
*RBM15::MRTFA*
*MNX1*
*KAT6Ar*
*ETS* family
*CBFB::MYH11*
*BCR::ABL1*
*CBFB*-GDXY
*UBTF*
*DEK::NUP214*
*GLISr*
*NPM1*
*GATA1*
*KMT2A-PTD*
*NUP98r*
APL
*PICALM::MLLT10*
*MECOM*
*HOXr*
*RUNX1::RUNX1T1*
*BCL11B*
*KMT2Ar*
Unclassified

*BCOR*  *STAG2*  *U2AF1*  *SRSF2*  *ZRSR2*  *EZH2*  *ASXL1*  *SF3B1*

P=0.001
P=0.007
P=0.04
P=0.001
P=0.006

**Extended Data Fig. 3 | See next page for caption.**

**Extended Data Fig. 3 | Transcriptional and mutational characterization of the study cohort. a**. UMAP (Uniform Manifold Approximation and Projection) plots and diffusion maps colored according to the WHO classification. **b**. A diffusion map colored according to molecular categories of the samples. DC: diffusion component, APL: acute promyelocytic leukemia. **c**. Expression of marker genes on UMAP plots and diffusion maps. Colors represent scaled expression levels. **d**. Cellular hierarchy scores inferred by CIBERSORT on UMAP plots and diffusion maps. Colors represent scaled scores. LSPC: leukemia stem and progenitor cell, GMP: granulocyte and macrophage projenitor, cDC: classic dendritic cell, ProMono: promonocyte, Mono: monocyte. **e**. UMAP plots and diffusion maps colored according to the French-American-British (FAB) classification **f**. A heatmap showing frequencies of defining gene alterations of AML, myelodysplasia-related in the WHO classification in each category. Colors denote the frequencies. Statistical significance was assessed by two-sided Fisher's exact test to calculate $P$ values of co-occurrence followed by the Benjamini-Hochberg adjustment for multiple testing to calculate $q$ values. No pair remained significant ($q < 0.05$) after adjustment, and $P$ values ($<0.05$) are shown instead.

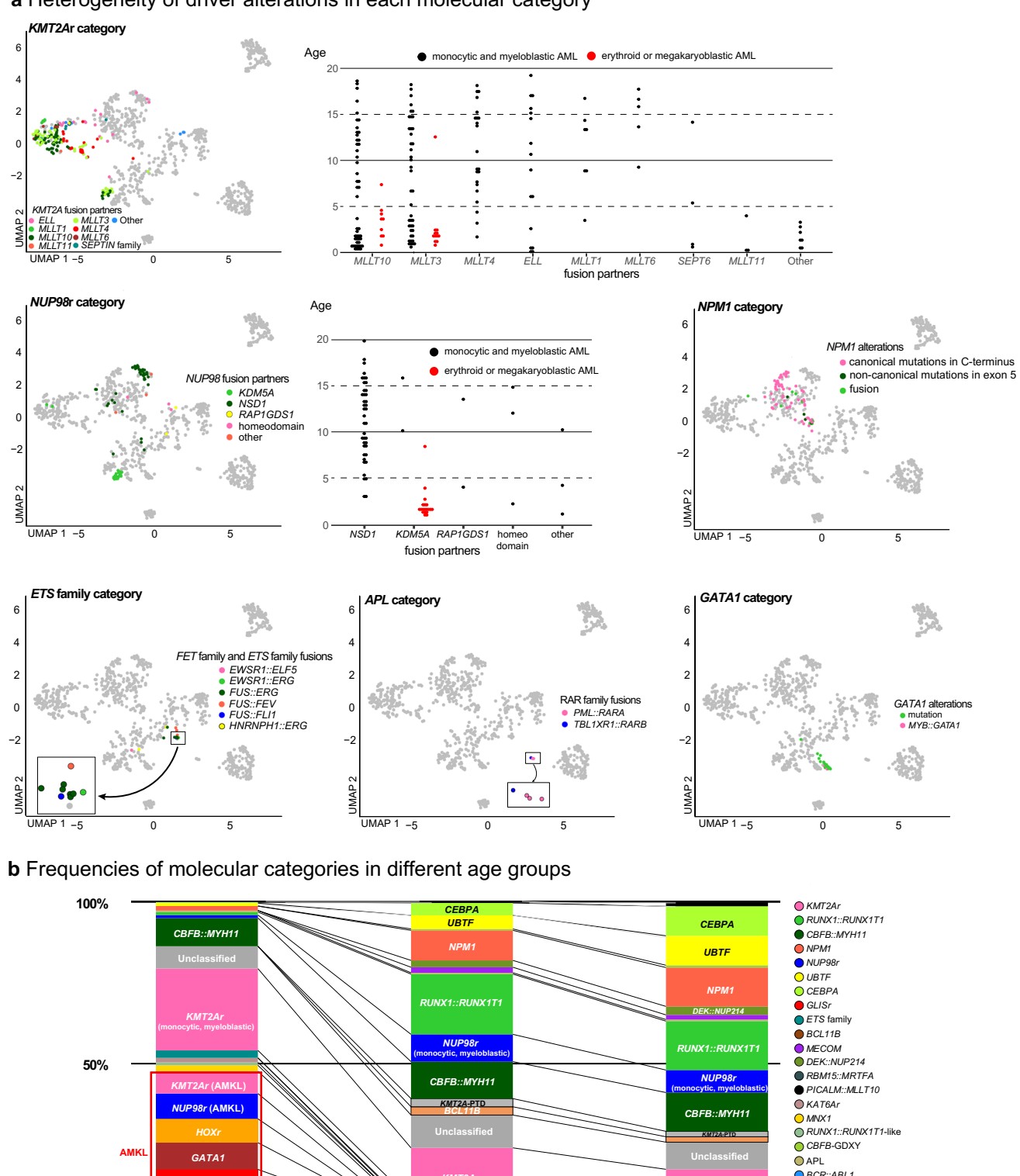

**Extended Data Fig. 4 | Details of molecular categories. a.** Details of molecular categories with multiple category-defining alterations. The distribution on the UMAP plot according to fusion partners (*KMT2A*r, *NUP98*r, *ETS* family, and APL categories) or mutation and fusions (*NPM1* and *GATA1* categories) are shown with colors representing the types of alterations. Age distributions according to fusion partners are also shown for *KMT2A*r and *NUP98*r. Acute megakaryocytic/erythroid leukemia (AMKL/AEL) cases are shown separately in red. **b.** Proportion of molecular categories among different age groups (**left**: age<3, **middle**: 3<age<10, **right**: 10<age). Each column is colored according to the molecular categories, and categories associated with AMKL/AEL phenotypes are highlighted in a red square. Representative category names are shown in the columns.

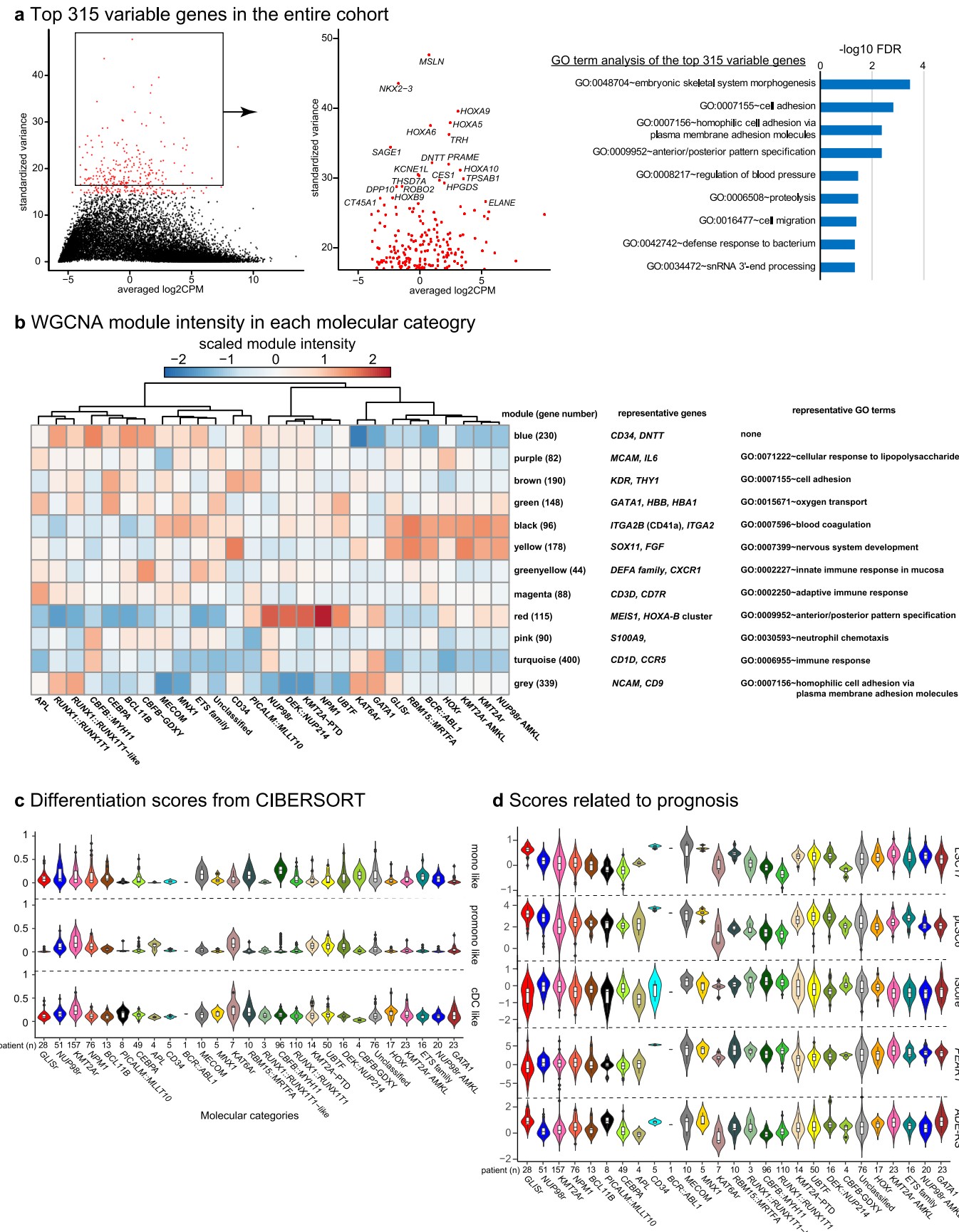

**a** Top 315 variable genes in the entire cohort

**b** WGCNA module intensity in each molecular cateogry

**c** Differentiation scores from CIBERSORT

**d** Scores related to prognosis

**Extended Data Fig. 5 | See next page for caption.**

**Extended Data Fig. 5 | Transcriptional analysis of the study cohort. a**. Plots showing averaged log2 CPM (count per million) values and standardized variance in the entire cohort (**left**). The top 315 variable genes used for the UMAP analysis were colored red, and representative variable gene names are shown in the right enlarged plot. The top results of the Gene Ontology (GO) term analysis by DAVID (Database for Annotation, Visualization and Integrated Discovery) are shown in the right panel. Bars represent logged FDR (false discovery rate<0.1). **b**. A heatmap colored according to scaled module intensities of WGCNA (weighed-gene correlation network analysis) in each molecular category. Representative genes and results of GO term analysis of genes in each module are shown on the right. Blue module enriched no GO term with FDR < 0.1. **c**. Distribution of differentiated cell-related hierarchy scores inferred by CIBERSORT among molecular categories. **d**. Distribution of prognostic scores among molecular categories. LSC17: leukemia stem cell 17 score, pLSC6: pediatric leukemia stem cell 6 score, iScore: inflammation-associated gene score, ADE-RS: Ara-C, Daunorubicin and Etoposide Drug Response Score. In **c** and **d**, lines of the box represent 25% quantile, median, and 75% quantile. The upper whisker represents the higher value of maxima or 1.5 x IQR, and the lower whisker represents the lower value of minima or 1.5 x IQR. Dots represent outliers. The colors of plots show molecular categories.

**a** Expression heatmap of *HOXA-B* cluster genes

**b** Expression and mutation patterns of *FLT3*, *NRAS* and *KRAS* in *HOXA-B* categories

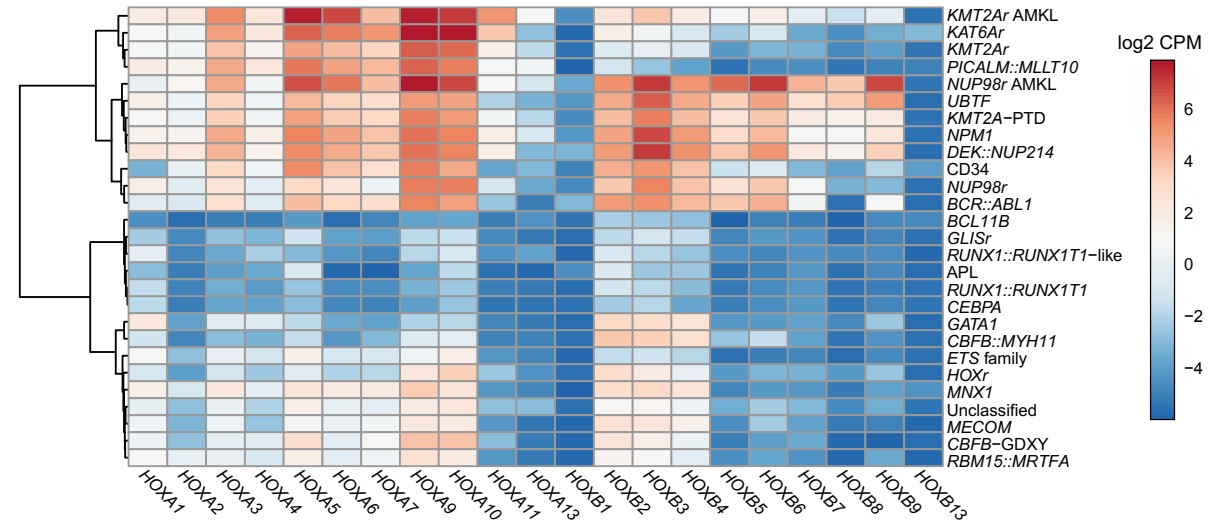

**Extended Data Fig. 6 | See next page for caption.**

**Extended Data Fig. 6 | Transcriptional and mutational characterization of clusters demarcated by *HOXA-B* expression. a**. A heatmap showing expression patterns of *HOXA* and *HOXB* cluster genes among molecular categories. Each panel color shows the expression level (log2CPM) of genes. Molecular categories are clustered using the Euclidean distance of the expression levels and the Ward method. **b**. Expression (**left**) and ProteinPaint of mutation patterns (**right**) of *FLT3* (**top**), *NRAS* (**middle**), and *KRAS* (**bottom**) in the *HOXA-B* categories. The distribution of log2CPM values among molecular categories is shown for the expression level, and the colors represent molecular categories. For the mutation plots, mutation types and frequencies in the *HOXA* and *HOXB* categories are shown separately, and the colors represent mutations types. Statistical significances of mutation distribution and frequency of each mutation were assessed by two-sided Fisher's exact test (*P* value), and no adjustment for multiple testing was applied. For each type of *NRAS* and *KRAS* mutations, variant allele frequencies (VAFs) are also shown. The lines of the box represent 25% quantile, median, and 75% quantile. The upper whisker represents the higher value of maxima or 1.5 x IQR, and the lower whisker represents the lower value of minima or 1.5 x IQR. Dots represent outliers.

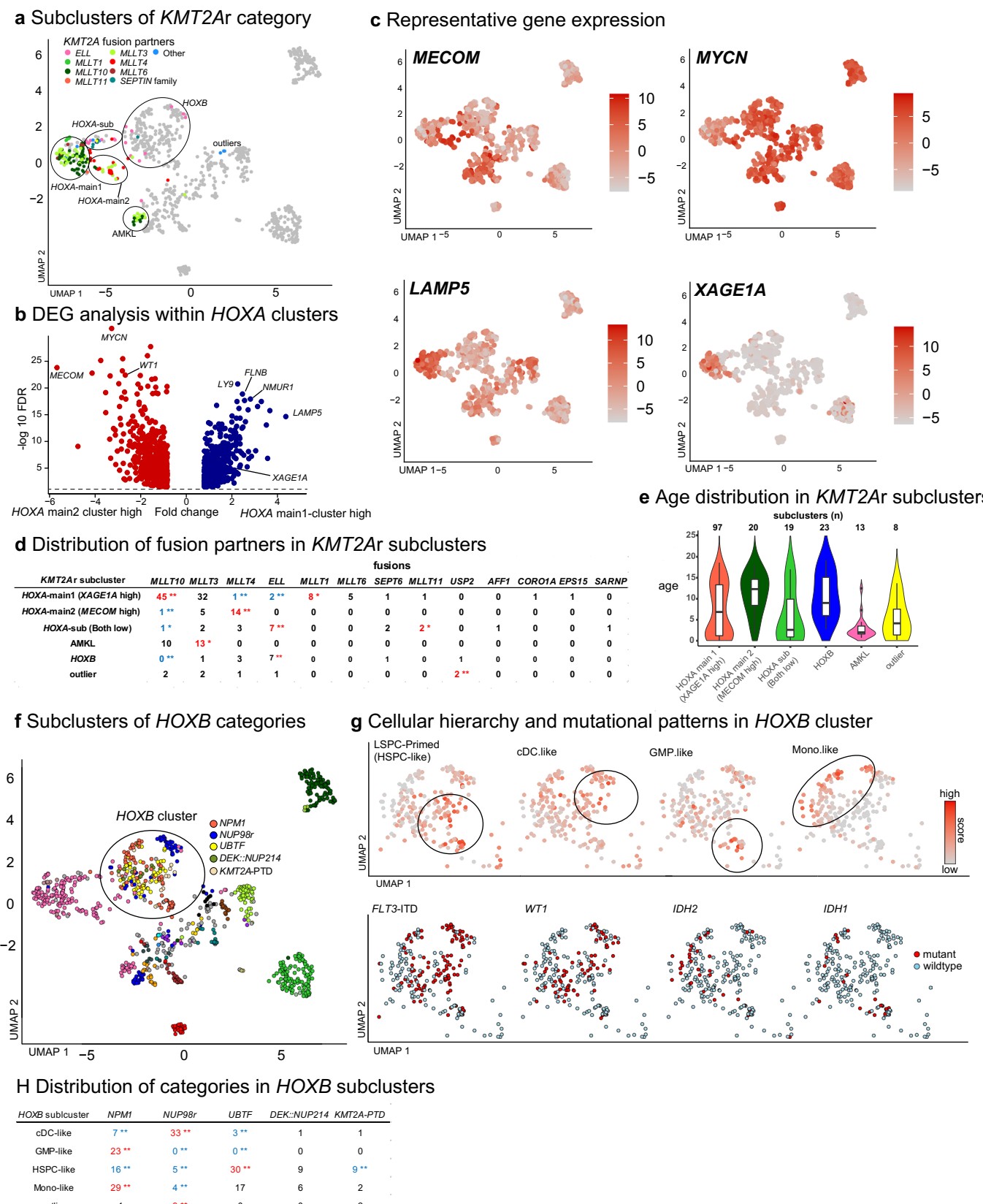

**a** Subclusters of *KMT2A*r category

**b** DEG analysis within *HOXA* clusters

**c** Representative gene expression

**d** Distribution of fusion partners in *KMT2A*r subclusters

**e** Age distribution in *KMT2A*r subclusters

**f** Subclusters of *HOXB* categories

**g** Cellular hierarchy and mutational patterns in *HOXB* cluster

**H** Distribution of categories in *HOXB* subclusters

Extended Data Fig. 7 | See next page for caption.

**Extended Data Fig. 7 | Molecular heterogeneity among *HOXA* and *HOXB* groups. a**. UMAP plot showing the distribution of fusion partners of *KMT2A*r among different clusters. The dot colors denote fusion partners. **b**. A volcano plot showing differentially expressed genes (DEG) between the *HOXA*-main1-2 clusters. Genes with absolute fold change > 2 and FDR < 0.05 are considered DEGs (red: *HOXA*-main2 cluster high, blue: *HOXA*-main1 cluster high). Representative gene names are shown. **c**. Expression of representative DEGs on UMAP plot. The dot colors represent the relative expression of the genes. **d**. The association of fusion partners of *KMT2A*r among different clusters. The statistical significance of the enrichment and exclusivity were assessed by two-sided Fisher's exact test followed by the Benjamini-Hochberg adjustment (*$P < 0.05$, **$q < 0.05$, blue: exclusive, red: enriched). **e**. Distribution of age at diagnosis among *KMT2A*r

different clusters. The colors of violin plots represent clusters and lines of the box represent 25% quantile, median, and 75% quantile. The upper whisker represents the higher value of maxima or 1.5 x IQR, and the lower whisker represents the lower value of minima or 1.5 x IQR. Dots represent outliers. **f**. UMAP plot highlighting molecular categories in the *HOXB* cluster. The dot colors denote molecular categories. **g**. Cellular hierarchy scores represented by the color (**top**) and patterns of frequent mutations (**bottom**) in the *HOXB* cluster. Circles in the top highlight clusters with high hierarchy scores. Blue and red dots in the bottom show mutational status. HSPC: hematopoietic stem and progenitor cell. **h**. The association of molecular categories and *HOXB* subclusters. The statistical significance of the enrichment and exclusivity were calculated and shown as in **d**.

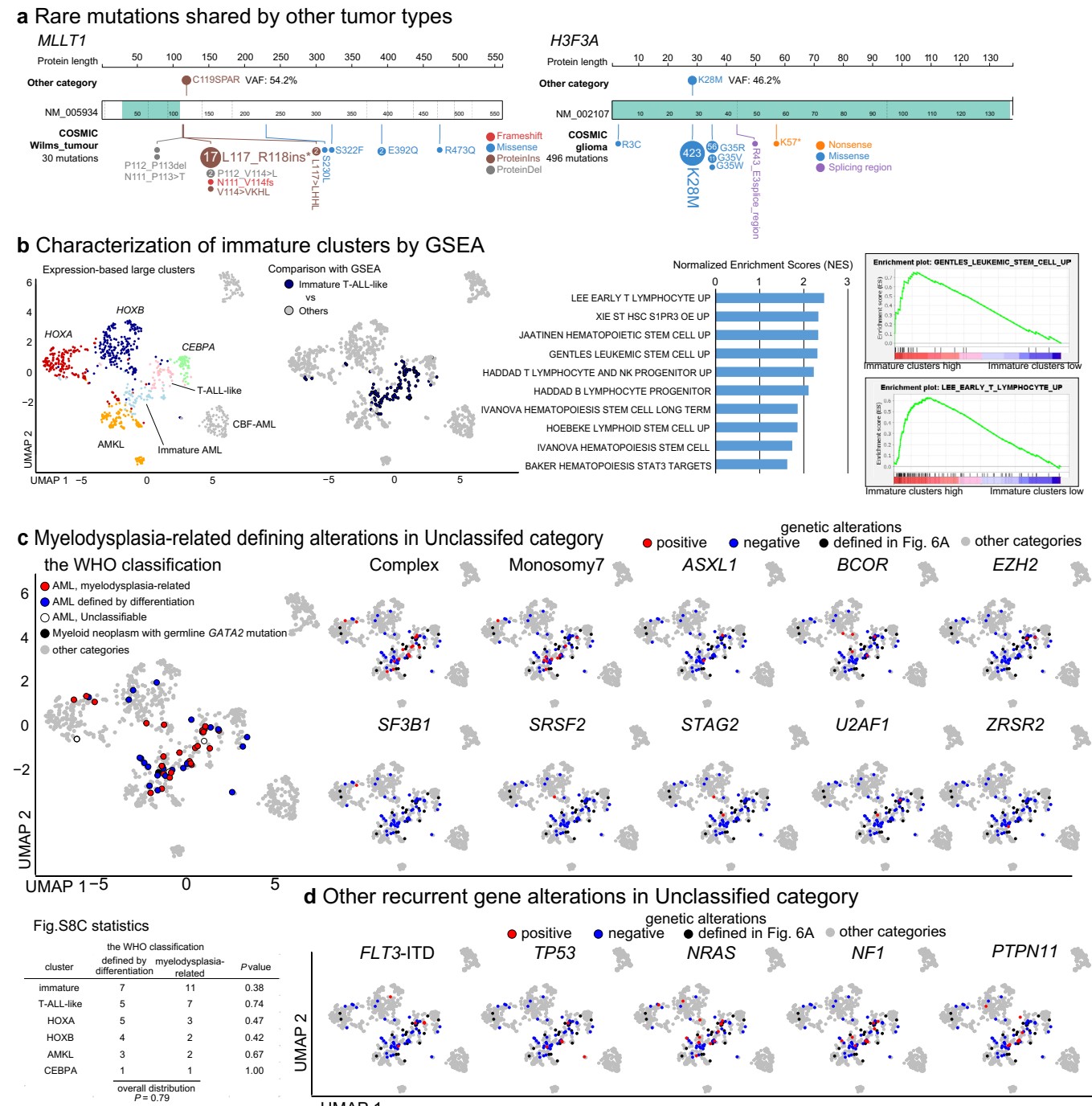

**Extended Data Fig. 8 | Characterization of cases without category-defining alterations. a**. ProteinPaint of rare somatic mutation in the study cohort. As comparisons, data from the COSMIC (Catalogue of Somatic Mutations in Cancer) database. Wilms tumor cohort for *MLLT1* mutation and glioma cohort for *H3F3A* are shown at the bottom. The colors represent mutation types. **b**. Design of GSEA (gene set enrichment analysis) comparing immature clusters with cluster membership 6, 9, and 16 with the rest of AML samples (**left**) and representative results for gene sets involved in hematopoietic stem cells or lymphocytes (**right**). Colors of dots of UMAP show clusters. Representative enrichment score plots are also shown. **c**. Distribution of the WHO classification (**left**) and myelodysplasia-related karyotypes and genetic alterations (**right**) in the Unclassified cases on UMAP plots. The dot colors of the right panel represent mutational status (red-positive, blue-negative), while black dots represent excluded Unclassified cases with recurrent alterations and gray dots represents other categories. The statistical significance of the enrichment and exclusivity of WHO classification and clusters were assessed by two-sided Fisher's exact test, and *P* values of cluster-wise comparison and overall distribution are shown in a table (**bottom**). **d**. Distribution of other recurrent genetic alterations in the Unclassified cases on UMAP plots. The dots are colored as in **c**.

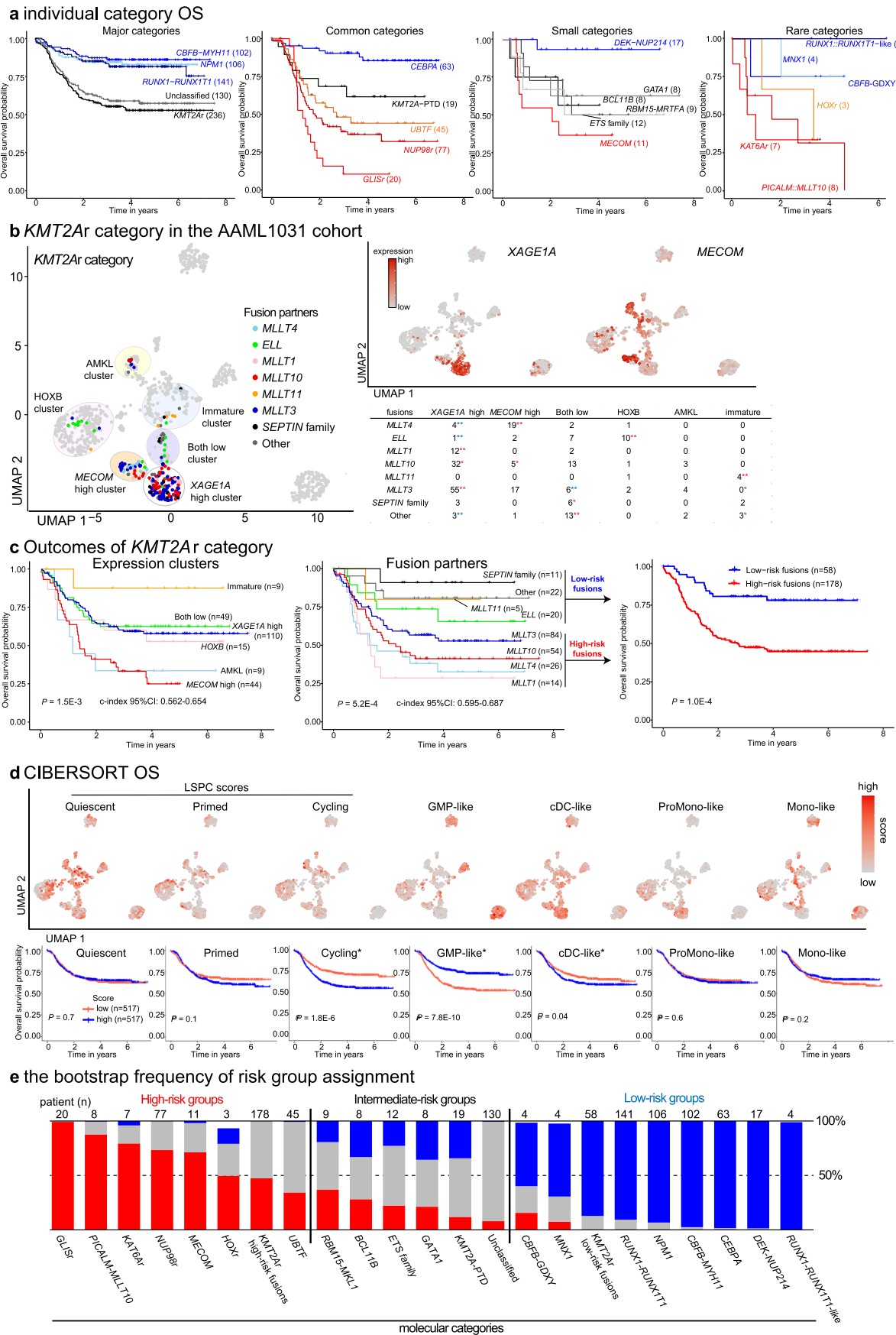

**a** individual category OS

**b** *KMT2A*r category in the AAML1031 cohort

**c** Outcomes of *KMT2A*r category

**d** CIBERSORT OS

**e** the bootstrap frequency of risk group assignment

**Extended Data Fig. 9 | See next page for caption.**

**Extended Data Fig. 9 | Clinical association of molecular categories and known prognostic factors in the AAML1031 cohort. a**. Kaplan-Meier curves of overall survival of patients in each molecular category. Category names and curves are colored according to outcomes (blue: favorable, black: intermediate, red: unfavorable). **b**. Details of *KMT2A*r category in the AAML1031 cohort showing the distribution of *KMT2A*r cases among transcriptional clusters colored by fusion partners (**left**) and by *XAGE1A* and *MECOM* expression (**top-right**) on UMAP plot, and the association of fusion partners of *KMT2A*r among different clusters (**bottom-right**). Circles on the UMAP highlight clusters (white: *XAGE1A* high, orange: *MECOM* high, purple: both low, pink: *HOXB*, yellow: AMKL, blue: immature). The statistical significance of the enrichment and exclusivity were assessed by two-sided Fisher's exact test followed by the Benjamini-Hochberg adjustment (*$P < 0.05$, **$q < 0.05$, blue: exclusive, red: enriched). **c**. Kaplan-Meier curves of overall survival of patients of *KMT2A*r with each fusion (**left**), in each cluster (**middle**), and Low and High-risk fusion groups by recursive partitioning (**right**). For the validity of prediction by *KMT2A*r fusion partners and clusters, c-index scores assessed by bootstrapping (1,000 times) were shown below the plots. **d**. Cellular hierarchy scores on UMAP plots (**top**) and Kaplan-Meier curves and statistical significance of overall survival (**bottom**). Significant scores in univariate analysis are highlighted with asterisks (Cycling, GMP-like, and cDC-like scores). For survival curves in **c-d**, statistical significance was assessed by the log-rank test, and *P* values are shown in the plots. **e**. Frequency of risk assignment by bootstrapping (1,000 times). Molecular categories are sorted according to the frequency within each risk group.

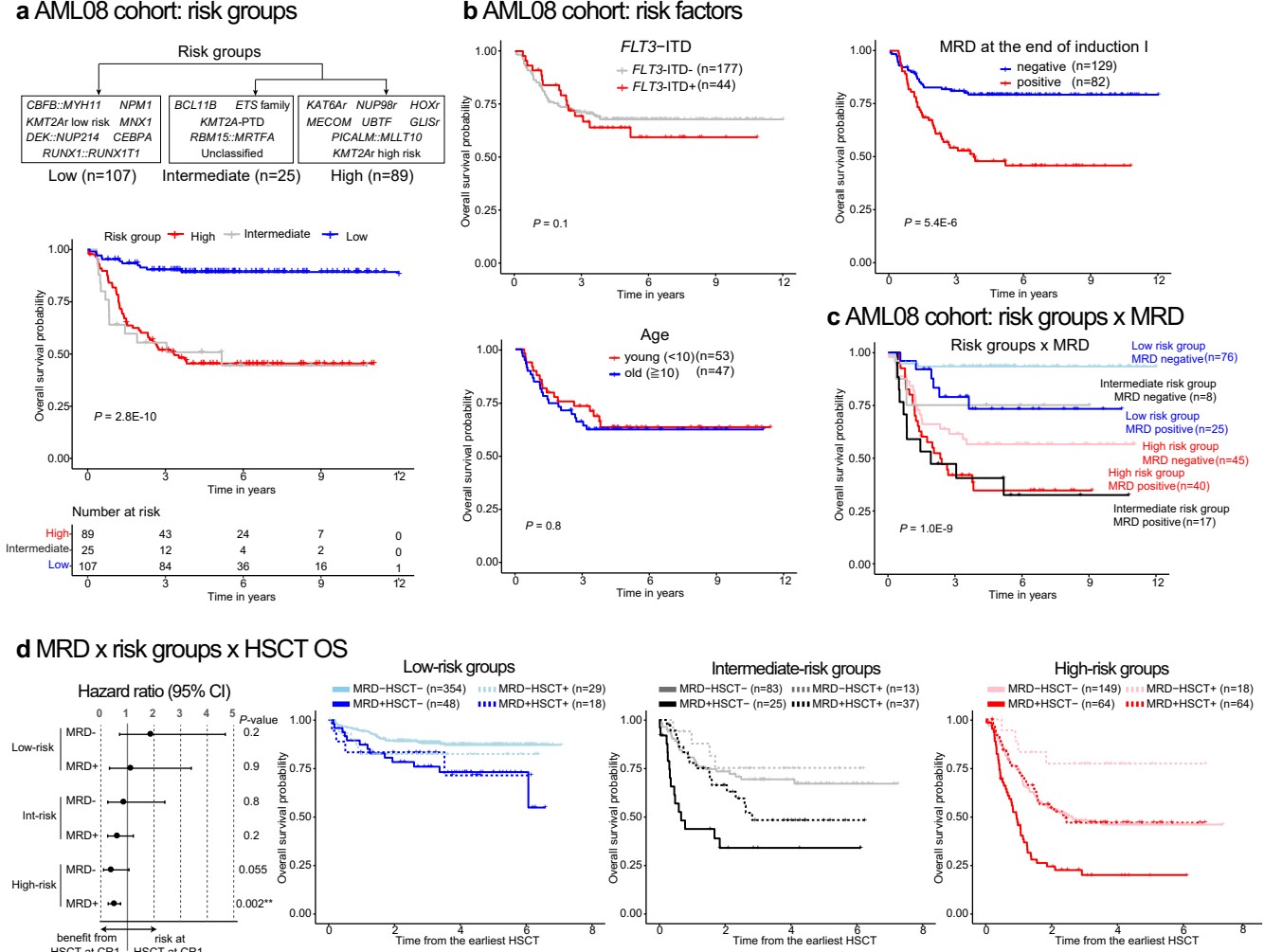

**Extended Data Fig. 10 | Validation of the prognostic model. a.** Grouping of molecular categories into Low, Intermediate, and High-risk groups (**top**) and Kaplan-Meier curves of overall survival of patients in each risk group (**bottom**) in the AML08 cohort. **b.** Kaplan-Meier curves and statistical significance of overall survival of patients with known prognostic factors (*FLT3*-ITD status: **top-left**, age: **bottom-left**, MRD (minimal residual disease) positivity at the end of the induction I: **top-right**) in the AML08 cohort. **C.** Kaplan-Meier curves of overall survival of patients in six risk strata using risk groups (Low-Intermediate-High) and MRD (measurable residual disease) positivity in the AML08 cohort. **d.** Outcomes in each risk group depending on MRD and HSCT (hematopoietic stem

cell transplant) status in the AAML1031 cohort. **left**-Hazard ratio (dot) and 95% confidence intervals (lines) in each group. **right**-Kaplan-Meier curves of overall survival. Survival curves start from the earliest transplant day within the cohort (day 96) and exclude patients who died before that timepoint. For survival curves in **a-c**, statistical significance was assessed by the log-rank test, and *P* values are shown in the plots. For **d**, the statistical significance of HSCT in each risk group was assessed by incorporating HSCT status as time-dependent variables and shown next to the hazard ratio plot. For survival analysis involving MRD status, patients with available MRD status (MRD+:n = 273, MRD-: n = 703) are included.

# Reporting Summary

## Statistics

For all statistical analyses, confirm that the following items are present in the figure legend, table legend, main text, or Methods section.

| n/a | Confirmed | |
|---|---|---|
| ☐ | ☒ | The exact sample size (*n*) for each experimental group/condition, given as a discrete number and unit of measurement |
| ☐ | ☒ | A statement on whether measurements were taken from distinct samples or whether the same sample was measured repeatedly |
| ☐ | ☒ | The statistical test(s) used AND whether they are one- or two-sided<br>*Only common tests should be described solely by name; describe more complex techniques in the Methods section.* |
| ☐ | ☒ | A description of all covariates tested |
| ☐ | ☒ | A description of any assumptions or corrections, such as tests of normality and adjustment for multiple comparisons |
| ☐ | ☒ | A full description of the statistical parameters including central tendency (e.g. means) or other basic estimates (e.g. regression coefficient) AND variation (e.g. standard deviation) or associated estimates of uncertainty (e.g. confidence intervals) |
| ☐ | ☒ | For null hypothesis testing, the test statistic (e.g. *F*, *t*, *r*) with confidence intervals, effect sizes, degrees of freedom and *P* value noted<br>*Give P values as exact values whenever suitable.* |
| ☒ | ☐ | For Bayesian analysis, information on the choice of priors and Markov chain Monte Carlo settings |
| ☒ | ☐ | For hierarchical and complex designs, identification of the appropriate level for tests and full reporting of outcomes |
| ☐ | ☒ | Estimates of effect sizes (e.g. Cohen's *d*, Pearson's *r*), indicating how they were calculated |

*Our web collection on statistics for biologists contains articles on many of the points above.*

## Software and code

Policy information about availability of computer code

| Data collection | HiSeq Control Software for Illumina HiSeq2000 and HiSeq2500 with most recent version at the time of sequencing were used to collect DNA and RNA sequencing data reported in this study. |
|---|---|
| Data analysis | StrongARM pipeline (not versioned), CICERO version.0.3.0, RNAseqCNV version 1.2.1, Bambino version1.07, RNAindel version3.0.4, VEP version95, PeCanPie (not versioned), BWA (WGS: v0.7.15-r1140 and v0.5.9-r26-dev; WES: v0.5.9-r26-dev and v0.5.9), Picard tools version 1.65, CREST version 1.0, CONSERTING (not versioned), Samtools version 1.16, VarScan2 version2.3.5, DNAcopy version 1.52.0, GRIN version 2, GISTIC version 2.0.23, gnomAD version 2.1.1, HTSeq version 0.11.2, Limma version 3.50.3, SVA version 3.42.0, Seurat version 4.1.0, destiny version 3.10.0, GSEA version 4.2.3, MSigDB gene sets c2.all version 7.5.1, WGCNA version 1.70-3, DAVID version 6.8, CIBERSORTx (not versioned), rpart version 4.1.19, survival 3.3.1, R version 4.0.2, pheatmap version 1.0.12, ggplot2 version 3.3.6, survminer version 0.4.9, iAdmix (not versioned) |

For manuscripts utilizing custom algorithms or software that are central to the research but not yet described in published literature, software must be made available to editors and reviewers. We strongly encourage code deposition in a community repository (e.g. GitHub). See the Nature Portfolio guidelines for submitting code & software for further information.

## Data

Policy information about availability of data

All manuscripts must include a data availability statement. This statement should provide the following information, where applicable:
- Accession codes, unique identifiers, or web links for publicly available datasets
- A description of any restrictions on data availability
- For clinical datasets or third party data, please ensure that the statement adheres to our policy

Genomic analyses in this study are based on the GENCODE GRCh37/hg19, and gnomAD version 2.1.1 was used for classification for germline and somatic mutations. The genomic data and expression data newly generated in this study (RNA-Seq: n=221, WGS: n=58, WES: n=7) have been deposited in the European Genome-Phenome Archive (EGA, RRID:SCR_004944), which is hosted by the European Bioinformatics Institute (EBI), under accession EGAS00001005760. Subsets of the new data (RNA-Seq: n=221, WGS: n=53, WES: n=5) have been also deposited to St. Jude Cloud under Pan-AML study (https://permalinks.stjude.cloud/panaml). Details are found in Supplementary Table 1.

For previously published RNA-Seq data (n=393), 266 are available either on EGA or St. Jude Cloud (1-9) or from the original publication (10). For the other 127 published cases (11), we downloaded the BAM files from EGA (EGAS00001004701). For previously published WGS data (n=198), 106 from the original publications (2, 3, 6-9) are available on either EGA or St. Jude Cloud, and the other 92 published BAM files (11) were downloaded from EGA (EGAS00001004701). For the previously published WES data (n=273), 153 with data from the original publications (1-9) are available either on St. Jude Cloud or EGA, and the BAM files for the other 120 published cases (11) were downloaded from EGA (EGAS00001004701).

We also downloaded data for publicly available but previously unpublished RNA-seq data (n=86) on St. Jude Cloud under the PCGP study (https://permalinks.stjude.cloud/permalinks/PCGP, n=8) and the RTCG study (https://platform.stjude.cloud/data/cohorts?dataset_accession=SJC-DS-1007, n=78). Similarly, we obtained unpublished WGS data (n=82: RTCG) and WES data (n=2: PCGP, n=99: RTCG study).

The data generated by the TARGET initiative (12,13) (n=187), including additional samples from the AAML1031 trial (14)(n=1034), are also available under accession phs000218 (TARGET-AML) and phs000465 (TARGET sub-study, data is available as a part of phs000218), managed by the NCI, and were obtained through GDC Portal managed by NCI under the TARGET-AML study (https://portal.gdc.cancer.gov/projects/TARGET-AML). Information about TARGET can be found at http://ocg.cancer.gov/programs/target. These sequencing data are available through controlled access as part of the NIH Genomic Data Sharing Policy (https://grants.nih.gov/grants/guide/notice-files/NOT-OD-14-124.html) and data access is restricted for academic use.

References
1. PMID: 30926971, 2. PMID: 34301788, 3. PMID: 30262806, 4. PMID: 29146900, 5. PMID: 25730765, 6. PMID: 27798625, 7. PMID: 35176137, 8. PMID: 28112737, 9. PMID: 23153540, 10. PMID: 31350825, 11. PMID: 34778799, 12. PMID: 29227476, 13. PMID: 3076086 14. PMID: 35349331

## Human research participants

Policy information about studies involving human research participants and Sex and Gender in Research.

| | |
|---|---|
| Reporting on sex and gender | Information on patient sex was based on biological features and collected along with other clinical data. Patient sex data was provided in supplemental table 1. We did not assume the impact of sex on the biological features or clinical outcomes of pediatric acute myeloid leukemia, and sex-focused analysis was not performed in this study. |
| Population characteristics | Patients had received a diagnosis of pediatric acute myeloid leukemia (AML) and the ages at diagnosis range from 0 to 23.5 (median 9.3). Of 881 patients with known sex, 418 patients (47.4%) were female and 463 patients were male. |
| Recruitment | Tumor samples from patients with acute myeloid leukemia from the St. Jude Children's Research Hospital tissue resource core facility were obtained with written informed consent. |
| Ethics oversight | St. Jude Children's Research Hospital institutional review board (IRB). |

Note that full information on the approval of the study protocol must also be provided in the manuscript.

# Field-specific reporting

Please select the one below that is the best fit for your research. If you are not sure, read the appropriate sections before making your selection.

☒ Life sciences          ☐ Behavioural & social sciences          ☐ Ecological, evolutionary & environmental sciences

For a reference copy of the document with all sections, see nature.com/documents/nr-reporting-summary-flat.pdf

# Life sciences study design

All studies must disclose on these points even when the disclosure is negative.

| | |
|---|---|
| Sample size | No sample size or power calculation was performed. The study cohort was determined by available patient samples with the diagnosis of AML |

| | |
|---|---|
| Sample size | and appropriate informed consent. We also included patients with available sequence data on public databases (St. Jude Cloud, EGA, and GDC data portal), to establish a large pediatric AML cohort of 887 patients fully characterized by sequence approaches. |
| Data exclusions | We excluded possibly duplicated samples estimated from pairwise genotype concordance comparison as well as low quality sequence data with possible tumor-normal contamination estimated from variant allele frequencies of somatic mutations as well as transcriptional analysis, which are not included in the final cohort of 887 patients. For patients with multiple time points at diagnosis or relapses, representative data points with good data quality with higher tumor purity estimation were included to establish a cohort with unique 887 patients. |
| Replication | For the genetic profiling of the study cohort, we performed the same analytical pipeline for an individual clinical study cohort of the AAML1031 study (n=1034), confirming the similar patterns of the overall molecular categories. For the clinical outcome data analysis, results from the AAML1031 study were validated using the AML08 study cohort (n=221, independent from the AAML1031 cohort, a part of this study cohort). |
| Randomization | No randomization of patients was performed in this study utilizing retrospective profiling of patients with available materials or sequence data. No analysis depending on patient background was performed in this study. |
| Blinding | No blinding was performed in the enrollment of patients or data collection of public data. Blinding in group allocation and in the following analyses were not possible as the grouping is based on the molecular characteristics of individual patients. |

# Reporting for specific materials, systems and methods

We require information from authors about some types of materials, experimental systems and methods used in many studies. Here, indicate whether each material, system or method listed is relevant to your study. If you are not sure if a list item applies to your research, read the appropriate section before selecting a response.

### Materials & experimental systems

| n/a | Involved in the study |
|---|---|
| ☐ | ☒ Antibodies |
| ☒ | ☐ Eukaryotic cell lines |
| ☒ | ☐ Palaeontology and archaeology |
| ☒ | ☐ Animals and other organisms |
| ☐ | ☒ Clinical data |
| ☒ | ☐ Dual use research of concern |

### Methods

| n/a | Involved in the study |
|---|---|
| ☒ | ☐ ChIP-seq |
| ☐ | ☒ Flow cytometry |
| ☒ | ☐ MRI-based neuroimaging |

## Antibodies

| | |
|---|---|
| Antibodies used | For the purification of the tumor population from patient samples, CD45dimCD33dim~positive population was sorted using the following antibodies. CD34 gating was added depending on the positivity of each patient sample.<br>CD45 PerCP-Cyanine5.5 (eBioscience cat# 8045-9459-120) 1:20<br>CD33 APC (eBioscience cat# 17-0338-42) 1:20<br>CD34 PE (Beckman cat# IM1459U) 1:5<br>Links:<br>CD45 PerCP-Cyanine5.5 (discontinued): https://www.thermofisher.com/order/catalog/product/8045-9459-120<br>CD33 APC: https://www.thermofisher.com/antibody/product/CD33-Antibody-clone-WM-53-WM53-Monoclonal/17-0338-42<br>CD34 PE: https://www.beckman.com/reagents/coulter-flow-cytometry/antibodies-and-kits/single-color-antibodies/cd34/im1459u |
| Validation | These antibodies were validated for detecting human proteins by the manufacturers using human peripheral blood mononuclear cells (CD45 and CD33) or KG1A cells (CD34). For each experiment, gating for tumor population (CD45dim x CD33dim~positive x CD34 variable) was confirmed using isotype controls. |

## Clinical data

Policy information about clinical studies

All manuscripts should comply with the ICMJE guidelines for publication of clinical research and a completed CONSORT checklist must be included with all submissions.

| | |
|---|---|
| Clinical trial registration | NCT00703820 (AML08) and NCT01371981 (the AAML1031 study); retrospective analysis only |
| Study protocol | https://clinicaltrials.gov/ct2/show/NCT01371981, https://clinicaltrials.gov/study/NCT00703820 |
| Data collection | AML08 trial was open from 8/2008 to 3/2017. Clinical and outcome data was obtained for AAML1031 trial from GDC data portal in June 2022. The original clinical trial started June 20, 2011, and the primary completion was March 31, 2019. |
| Outcomes | AML08 data was obtained from study PI and co-author Dr. Jeffrey Rubnitz and from PMID 31246522. Publicly available data for AAML1031 was obtained from GDC data portal was analyzed retrospectively as described in the manuscript. |

# Flow Cytometry

## Plots

Confirm that:

☒ The axis labels state the marker and fluorochrome used (e.g. CD4-FITC).

☒ The axis scales are clearly visible. Include numbers along axes only for bottom left plot of group (a 'group' is an analysis of identical markers).

☒ All plots are contour plots with outliers or pseudocolor plots.

☒ A numerical value for number of cells or percentage (with statistics) is provided.

## Methodology

| | |
|---|---|
| Sample preparation | For patients with less than 60% blasts, cryopreserved patient samples (bone marrow, peripheral blood) were thawed in IMDM media containing 20% FBS and subjected to flow-sorting for the tumor population before sequencing. |
| Instrument | Cell sorting was performed  using a FACSAria III instrument (BD Biosciences) |
| Software | FACSDiva 9.0 software (BD Biosciences) was used for data collection and gating for sorting. |
| Cell population abundance | Enrichment of the tumor population was confirmed flow cytometric analysis of the post-sorting samples (generally > 90%). |
| Gating strategy | Live cells were first gated using FSC-A and DAPI (BD cat# 564907) gating, followed by singlet gating (SSC-W x FSC-A). The myeloid population was further gated as CD45 dim x FSC variable population. CD34 gating for the blast population was considered depending on the positivity of the tumor population in each patient. |

☒ Tick this box to confirm that a figure exemplifying the gating strategy is provided in the Supplementary Information.

