## [Peer Review File · Nature Genetics]

Peer Review Information

Manuscript Title: A new genomic framework to categorize pediatric acute myeloid leukemia

Corresponding author name(s): Dr Jeffery (M) Klco

Reviewer Comments & Decisions:

Decision Letter, initial version:

27th Jun 2023

Dear Dr Klco,

Your Article, "Proposal of a new genomic framework for categorization of pediatric acute myeloid leukemia associated with prognosis" has now been seen by 3 referees. You will see from their comments below that while they find your work of interest, some important points are raised. We are interested in the possibility of publishing your study in Nature Genetics, but would like to consider your response to these concerns in the form of a revised manuscript before we make a final decision on publication.

Reviewers #1 and #3 are broadly positive about the work. Both have made relatively minor technical comments, and requests to improve overall clarity, positioning, and presentation. Reviewer #2 has clinical expertise and has reservations about the overall clinical/mechanistic impact of the study and we'd encourage you to bear these comments in mind as you prepare your revision.

We therefore invite you to revise your manuscript taking into account all reviewer and editor comments. Please highlight all changes in the manuscript text file. At this stage we will need you to upload a copy of the manuscript in MS Word .docx or similar editable format.

*2) If you have not done so already please begin to revise your manuscript so that it conforms to our

Article format instructions, available

[here](http://www.nature.com/ng/authors/article_types/index.html).

*3) Include a revised version of any required Reporting Summary:

[redacted]

We hope to receive your revised manuscript within four to eight weeks. If you cannot send it within this time, please let us know.

Sincerely,

Safia Danovi
Editor
Nature Genetics

Referee expertise:

Referee #1: AML, transcriptomics

Referee #2: paediatric AML, clinical, genomics

Referee #3: AML genomics, clinical

Reviewers' Comments:

Reviewer #1:

Remarks to the Author:

Adult AML is a highly studied poor prognosis disease, especially in elderly people. Pediatric AML (pAML) is less well studied, although past work has suggested the distribution of driver mutations differs compared to adult AML. For better risk stratification of pAML, the authors profiled the transcriptome 895 pAML cases, including both diagnostic and relapse cases. For 671 of these cases, they also had either exome or whole genome sequencing. They identified all the major WHO categories of pAML, and then went on to explore new categories in detail, which included identifying significant co-occurrence of mutations. They also used the RNA-seq to identify transcriptional signatures among the different categories of patients, and found that HOXA and HOXB expression could be used to identify distinct mutational categories of patients. They then went on to use their RNA-seq analysis pipeline to explore a previously published pAML analysis (AAML 1031), and associate the different patient clusters to specific mutational subsets with distinct clinical outcomes. In short, they have provided a new way to risk stratify pAML patients based on identifying mutations and transcriptional cohorts using RNA-seq analysis.

By its very nature, the work is inherently descriptive. That said, it is an impressive collection of data and a comprehensive analysis. This work will provide an important resource for the field, as well as being potentially clinically important.

I only have a few minor concerns:

1. Some of the numbers in Figure 1 E and F (or in the results text) don't match. For example, Fig 1E indicates that there were 77 NPM1 mutant samples, Fig 1F suggests 67, while the text in the results claims 68 patients. Similarly, Fig 1 E indicates 195 KMT2A alterations, while 1F indicates 180 KMT2A gene rearrangements. Is there a simple explanation for these discrepancies?
2. What is the main difference between Figure 2B and Figure S2? They seem to contradict each other slightly and indicate some differences in which mutations co-occur.

Reviewer #2:

Remarks to the Author:

In this manuscript, Umeda, et al. conduct a genomic and transcriptomic profiling of pediatric AML datasets. The authors explore the clinical and prognostic significance of genomic categories. Overall, the quality of the analysis is high and the paper is well written and nicely collates and validates the clinical and prognostic impact of genomic lesions in pAML. While the genomic characterization is comprehensive - uncovering a few potential new driver lesions, defining mutational co-occurrence patterns in more granular detail than previous works, and identifying two major expression signatures - the work is overall more hypothesis generating than mechanistic. Additionally, while providing some potential new prognostic categories, in general, the proposed framework is largely similar to current risk stratification of pediatric AML (genomic classifiers + MRD). Thus, while the paper covers a breadth of topics, the depth of the analysis and immediate potential to impact patient care is lacking.

1. A lot of the focus is contrasting the findings with classification used for adult AML. While it's an interesting comparison, I feel it already well established that pAML genomics, clinical features and

outcomes are divergent from adults. It would be more valuable to better understand how the data presented here could potentially impact how pAML is currently risk stratified (does this improve upon how large consortia such as BFM, COG, SJCRH, DFCI, etc) risk classify patients.

2. While there is some integration of genomic classifiers and MRD, can the authors identify a group in which one better discriminates outcome. For instance, are the typically good prognosis genomic lesions that MRD would define patients who would benefit from HSCT, or poor prognosis lesions who may be well if MRD-?
3. How does HOX expression pattern fit into risk stratification. It seems to largely align with genomic classifiers, so its not entirely clear how it would potentially come into play for better prognostication.
4. Data supports that not all KMT2Ar impact prognosis similarly (1;11 is good prognosis, 6;11 is poor, etc). Thus, it is not surprising that KMT2Ar AMLs fall into an intermediate category. Why did the authors not incorporate specific KMT2Ar into their prognostication schema?
5. Of course, outcomes from retrospective data are contingent upon the therapy delivers. Can the authors comment on the potential impact of the risk stratification and therapy used in the AAML1031 study? For example, did FLT3ITD not have a prognostic impact because FLT3 inhibitor was used for those with high AR FLT3ITD? Did HSCT in CR1 potentially impact outcome in those who were assigned it based on risk classification?
6. The authors state the enrichment for FLT3ITD in the HOXB category implies 'data-driven implementation of FLT3 inhibitor to HOXB subtypes can be effective'. It is not clear what is meant by this? Are they arguing patients in the HOXB cluster should get FLT3 inhibitors or that identification of HOXB status should inform which FLT3ITD patients should get FLT3 inhibitors? And how would this improve upon simply giving FLT3i to all patients with FLT3ITD lesions (or high AR ones at least)?

Minor:

7. How did was somatic versus germline status confirmed?
8. Why in figure 2D are NUP98r and NPM1mut crossed over (likewise, CEPBA and UBTF)? It gives the false impression that there are more major differences between the left and right sides of the figure. These major categories that are essentially the same should just be aligned as the top 3 are.

Reviewer #3:

Remarks to the Author:

Umeda and colleagues assembled a cohort of 895 pediatric cases, performed RNA-seq on all and assembled WGS/WES data on 75% of cases, and defined new molecular subgroups with distinct clinical features. Comments:

1. Given the rarity of pediatric AML and the diverse origin of the samples collected here, it would be worth proving that the 895 cases are indeed "unique," as stated on p3. This should be feasible using a small number of polymorphic fingerprinting markers extracted from NGS data. Furthermore, it would be important to state somewhere that there is no overlap between this cohort and the AAML1031 AML08 cohorts used for validation.
2. The race and ethnicity of study participants should be reported, either using self-reported information or (better yet) using ancestry-informative markers from NGS data.
3. I was surprised to note that mutations in TP53, ASXL1, and JAK2 do not appear in Fig 1C. This figure panel appears to be a subset of the larger gene list in Fig S2, but the criteria for inclusion in Fig 1C are not explained.
4. Calling SNVs/Indels from RNA-seq data is challenging...this is compounded here by the absence of a matched normal control. Similarly, a fraction of WES/WGS samples lacked a germline control. 75% of

cases have both RNA-seq and WGS/WES data. This could provide some level of cross-platform validation, but that was not described. The manuscript does not appear to report novel somatic variants which tempers this concern somewhat. Still, some orthogonal validation of calls made only by RNA-seq appears warranted. This could be addressed by sequencing remission samples or matched normal tissue, if available.

5. Conversely, the lack of matched normal sequencing data from many patients could have led to an underestimate of the number of cases with inherited predisposition drivers. Indeed, the frequency reported here is lower than expected.

6. The organization of categories in Fig 2D gives the false impression that many patients in genetically-defined subgroups from WHO are reassigned to new categories in the proposed molecular framework, whereas this is just a consequence of the way the categories are ordered. NPM1, NPM1r, and CEBPA should be presented in the same order in both systems. This will fix the problem and allow the new entities to stand out (e.g., UBTF, GLIS4, GATA1). The last sentence in this section on p8 should emphasize that with the identification of 12 new molecular categories, the new classification system captures 91.4% of cases, up from 68.3% by WHO.

7. The largest fraction of cases assigned to the Unclassified category in the molecular system were classified as myelodysplasia-related in WHO. This should be noted in the section on Unclassified cases (and perhaps retained as a non-molecularly defined category since they have distinct clinical features). In adults, many of these cases would harbor splicing gene mutations. These mutations are less common in pAML, although it is worth noting that one of the most frequently mutated splicing genes (SRSF2) was not included in the 86 gene panel.

8. The clinical experience with a single Menin inhibitor is mentioned in the discussion. To avoid the perception of bias, this discussion should be broadened slightly since there are now several compounds in the clinic with preliminary results reported.

Minor:

There are numerous callouts to figure panels that are discordant, presumably because text was not updated after figures were changed.

Author Rebuttal to Initial comments

We thank the Editorial staff and the 3 Reviewers for their helpful discussions and comments regarding our manuscript. We have been able to successfully address the reviewer's comments and we feel that our overall manuscript has been significantly improved. Below is a point-by-point response to each comment.

Reviewer #1:

1. Some of the numbers in Figure 1 E and F (or in the results text) don't match. For example, Fig 1E indicates that there were 77 NPM1 mutant samples, Fig 1F suggests 67, while the text in the results claims 68 patients. Similarly, Fig 1 E indicates 195 KMT2A alterations, while 1F indicates 180 KMT2A gene rearrangements. Is there a simple explanation for these discrepancies?

For *NPM1* mutations and categories, the WHO classification defines acute myeloid leukemia with *NPM1*

mutations (n=66 in the updated figure). Our classification includes *NPM1* fusions (n=10) as *NPM1* category (n=76) based on the findings from a recent report¹ showing that *NPM1* fusions are clinically and functionally similar to *NPM1* mutations. Our data also showed that AMLs with *NPM1* fusions are transcriptionally comparable to AML with *NPM1* mutations (Fig.S4A).

Also, in the updated Fig.1E, we separately show *KMT2A* rearrangement (n=180) and *KMT2A*-PTD (N=14), which define different categories to avoid confusion. One case with *KMT2A* rearrangement has germline *ETV6* mutation and was assigned WHO classification as “AML with germline predisposition”, resulting in 179 cases of AML with *KMT2A* rearrangements in the WHO classification in Fig.1F.

2. What is the main difference between Figure 2B and Figure S2? They seem to contradict each other slightly and indicate some differences in which mutations co-occur.

We intend Fig.2B to show that all cases in each molecular category have defining gene alterations, and the colors of each panel show the percentages of cases with the gene alterations. Fig.S2 is designed to show pairwise co-occurrence and exclusivity of altered genes not limited to defining alterations with statistics (such as an exclusivity between *NRAS* and *FLT3*-ITD or co-occurrence of complex karyotype and *HOX* cluster alterations).

Reviewer #2:

1. A lot of the focus is contrasting the findings with classification used for adult AML. While it's an interesting comparison, I feel it already well established that pAML genomics, clinical features and outcomes are divergent from adults. It would be more valuable to better understand how the data presented here could potentially impact how pAML is currently risk stratified (does this improve upon how large consortia such as BFM, COG, SJCRH, DFCl, etc) risk classify patients.

We thank the reviewer for suggesting an additional analysis to emphasize the clinical impact of the proposed molecular category and risk stratification framework. We assigned multiple risk stratifications currently used in clinical trial groups of pediatric AML (BFM, COG AAML1031, SJCRH AML16, JPLSG) and European LeukemiaNet (ELN) as representative risk stratification for adult AML and compared the predictive values with that of our framework.

To quantify the predictiveness of each risk stratification, we computed Harrell's concordance index² for a Cox model with risk group as the sole predictor of EFS and/or OS (Fig.S10E). We then used a bootstrap procedure to evaluate the variability of the concordance index estimate for each risk stratification. We generated 1,000 bootstrap data sets by resampling patients with replacement. We then computed the concordance index of each risk stratification for each bootstrap data set. To compare each pair of risk stratification systems, we computed the difference between the concordance index values for each bootstrap data set and then found the 95% bootstrap confidence interval for the difference of the concordance indices. For our three-branch survival tree risk stratification, we repeated the determination of the three-branch survival tree for each bootstrap data set to evaluate the stability of assignment of molecular groups to risk groups and account for this variability in comparing it with previously defined risk stratifications.

For validation, we classified participants of the AML08 clinical trial according to the three-branch survival tree fit to AAML1031 data and the other risk stratification criteria. We then computed the concordance index in a bootstrap procedure to quantify and compare the risk stratifications. The bootstrap procedure was the same as for AAML1031 data, except the three-branch tree definition remained fixed in the validation analysis of the AML08 data. The results showed that the risk classification based on the three-branch survival regression tree model developed on the AAML1031 cohort was among the most predictive risk classification systems in the validation evaluation on the AML08 cohort.

2. While there is some integration of genomic classifiers and MRD, can the authors identify a group in which one better discriminates outcome. For instance, are the typically good prognosis genomic lesions that MRD would define patients who would benefit from HSCT, or poor prognosis lesions who may be well if MRD-?

We appreciate this question raising a great point for the discussion. In the initial submission, we could not address how HSCT affected the outcome due to lack of data regarding the timing of HSCT in the first remission (CR1). In this revised manuscript, we obtained the timing of HSCT in collaboration with COG biostatistician Todd Alonzo (now added as a co-author) and incorporated it as a variable to address the effect of HSCT in different risk groups. We reported Kaplan-Meier estimates of overall survival from the time of the earliest transplant in the AAML1031 study. We also fit Cox models to estimate the benefit of transplant (as a time-dependent covariate) for each patient subgroup defined by risk and MRD status. Our analysis shows statistically significant evidence that transplant is beneficial for high-risk MRD+ patients; it also estimates a substantial (but not statistically significant) benefit of transplant for MRD- high-risk patients and MRD+ intermediate risk patients. There is no evidence that transplant is beneficial for MRD- intermediate risk patients or low-risk patients.

3. How does HOX expression pattern fit into risk stratification. It seems to largely align with genomic classifiers, so its not entirely clear how it would potentially come into play for better prognostication.

We propose HOXA and HOXB categories because of their similar expression profiles and potentially shared biological mechanisms for the development of pAML, such as KMT2A-Menin complex in both HOXA and HOXB categories, to discuss possible molecular target therapies. While the outcomes of patients with AMLs in the HOXA and HOXB categories are generally poor, the outcomes of *NPM1* AML is commonly favorable and it is confirmed in this manuscript, suggesting that HOX gene expression alone does not define AML with unfavorable outcomes and that outcomes need to be discussed in the context of driver gene alterations and cooperating mutations.

4. Data supports that not all KMT2Ar impact prognosis similarly (1;11 is good prognosis, 6;11 is poor, etc). Thus, it is not surprising that KMT2Ar AMLs fall into an intermediate category. Why did the authors not incorporate specific KMT2Ar into their prognostication schema?

Thank you for raising a great point to improve our model. Although we intended to focus on molecular category to simplify the framework in the initial submission, given that risk-stratification according to fusion partners of *KMT2A* rearrangements are broadly accepted (e.g., BFM, JPLSG, and ELN), we incorporated

KMT2A fusion partners in the risk-stratification model in the revised manuscript (Fig.S9C, Fig6D).

We first fit a survival regression tree model to the outcomes of KMT2Ar patients to group fusions into KMT2Ar high-risk and KMT2Ar low-risk groups. We then used these groups in a second survival regression tree model fit to the outcomes of all patients to derive our three-group risk classification. The KMT2Ar high and low risk subgroups were assigned to different branches in the final three-group survival regression tree model fit to all patients. We feel this greatly improved the value of our three-group risk group assignment.

5. Of course, outcomes from retrospective data are contingent upon the therapy delivers. Can the authors comment on the potential impact of the risk stratification and therapy used in the AAML1031 study? For example, did FLT3ITD not have a prognostic impact because FLT3 inhibitor was used for those with high AR FLT3ITD? Did HSCT in CR1 potentially impact outcome in those who were assigned it based on risk classification?

We appreciate this comment about potential impact of treatments on outcomes. In the AAML1031 study, a subset of *FLT3*-ITD AR high patients (arm C) were intended to receive sorafenib treatment and HSCT in CR1. Comparison with *FLT3*-ITD HR patients in different treatment arm (arm B and C) confirmed the therapeutic effect of sorafenib for this subset in the AAML1031 study³. Data about which patient received sorafenib treatment is not publicly available, and we were unable to answer directly to this question, while the trend toward longer EFS and comparable OS of *FLT3*-ITD HR patients could indicate the effect of sorafenib treatment. This could be also due to the impact on HSCT, which showed a trend of better outcome among High-risk and Intermediate risk with MRD as shown in the answer to comment #2-2. Our validation cohort also utilized sorafenib for *FLT3*-ITD+ cases independent of allelic status and thus is not optimal to assess this question.

In the manuscript, we describe the results with possible explanation as follows.

“Univariate analyses revealed that age and *FLT3*-ITD were not prognostic, which could reflect sorafenib given to *FLT3*-ITD high allele ratio patients in the AAML1031 study (Fig.6E, Fig.S9D).”

6. The authors state the enrichment for FLT3ITD in the HOXB category implies ‘data-driven implementation of FLT3 inhibitor to HOXB subtypes can be effective’. It is not clear what is meant by this? Are they arguing patients in the HOXB cluster should get FLT3 inhibitors or that identification of HOXB status should inform which FLT3ITD patients should get FLT3 inhibitors? And how would this improve upon simply giving FLT3i to all patients with FLT3ITD lesions (or high AR ones at least)?

From the expression profiles and mutation pattern, we hypothesize that inhibition of *FLT3* wildtype in AML in HOXB categories without *FLT3*-ITD could be effective. This concept aligns with a clinical trial testing quizartinib treatment for *FLT3*-ITD negative adult AML (NCT04107727). However, clinical data presented in this manuscript does not directly support this hypothesis, and we modified the description as follows in accordance with the answer to #2-5.

“Also, the high frequency of *FLT3*-ITD in categories with HOXB expression implies that *FLT3* signaling is closely related to the biology and that treatment with *FLT3* inhibitors to *FLT3*-ITD+ HOXB subtypes independent of the allelic ratio may be effective.”

Minor:

7. How did was somatic versus germline status confirmed?

When applicable, somatic and germline mutations were called separately as described in the Method section. For somatic mutation calls from DNA samples with germline controls, germline variants were filtered out if present in the matched germline sample. For mutation calls from DNA samples without germline controls, possible germline variants were filtered by excluding variants with gnomAD population allele frequency >0.1%.

For mutation calling from RNA-seq, candidate SNVs/Indels were called by Bambino or RNAIndel, annotated by VEP, filtered by excluding variants with gnomAD population allele frequency >0.1% as possible germline variants. Candidate variants were considered germline or artifacts if present in >5% of the cases. Furthermore, for Indel calls, RNAIndel uses a machine learning method to classify variants into somatic, germline and artifact. Finally, these somatic mutations were subjected to PeCanPie/MedalCeremony to assign pathogenicity labels, and only pathogenic and likely pathogenic mutations were used in the downstream analyses, collectively ensuring exclusion of germline mutations from the RNA-seq pipeline.

For germline mutation calling, we focused on cases with tumor-germline paired samples and 15 genes whose germline mutations are used for WHO classification (Table.S25). Possible germline mutations were subjected to computational filtering and review by a variant scientist based on recommendations from the American College of Medical Genetics and Genomics and the Association for Molecular Pathology and the Clinical Genome Resource as described in the Method section.

8. Why in figure 2D are NUP98r and NPM1mut crossed over (likewise, CEPBA and UBTF)? It gives the false impression that there are more major differences between the left and right sides of the figure. These major categories that are essentially the same should just be aligned as the top 3 are.

We intended the figure to emphasize the frequent categories in each classification system at the top in the initial submission. Due to the inclusion of *NPM1* fusions into the *NPM1* category, *NPM1* categories are more frequent than *NUP98r*, and they crossed between the WHO classification and molecular categories. For *UBTF* category, they are not specifically defined in the WHO classification but more frequent than *CEBPA* category, so the orders cross between these two classifications. According to this comment and Reviewer #3 comment 6, we reordered molecular category to show categories defined by the WHO classification first followed by newly proposed categories in the revised manuscript.

Reviewer #3:

1. Given the rarity of pediatric AML and the diverse origin of the samples collected here, it would be worth proving that the 895 cases are indeed “unique,” as stated on p3. This should be feasible using a small number of polymorphic fingerprinting markers extracted from NGS data. Furthermore, it would be important to state somewhere that there is no overlap between this cohort and the AAML1031 AML08 cohorts used for validation.

We thank the reviewer for suggesting the potential duplicates among pediatric AML patients from various data source. We performed genotype fingerprinting using SNPs with ≥ 20 coverage in RNA-Seq BAM files. Although we purposely included unique patients with different sample/patient IDs obtained from the St Jude Biorepository in a coded manner in the initial submission, pairwise comparison among 895 patients in the initial submission found 8 pairs (despite unique IDs) with concordance of SNPs higher than 90%, which are most likely to be duplicates according to manual inspection of SNPs and available clinical records. We kept one representative samples from these 8 duplicated pairs, resulting in the updated study cohort of 887 unique pediatric AML patients. We apologize for this initial oversight. We also included a comment that the AML08 cohort is independent from the AAML1031 cohort.

2. The race and ethnicity of study participants should be reported, either using self-reported information or (better yet) using ancestry-informative markers from NGS data.

As not all patients had self-reported information of ethnicity or race, we performed inference of ancestry and race using iAdmix program using only coding SNPs⁴. Results from iAdmix annotated 57.4% as White, 13.9% as Hispanic, and 12.1% as Black in this study cohort, and individual information was included in Table. S1.

3. I was surprised to note that mutations in TP53, ASXL1, and JAK2 do not appear in Fig 1C. This figure panel appears to be a subset of the larger gene list in Fig S2, but the criteria for inclusion in Fig 1C are not explained.

Due the limited space, we included genes with 19 or more mutations in Fig.1C in the initial manuscript, while *TP53* (n=18), *ASXL1* (n=15), *JAK2* (n=13) were excluded from the figure. We adjusted the size of the characters to include genes with 15 or more mutations in the revised figures and described the threshold in the figure legend.

4. Calling SNVs/Indels from RNA-seq data is challenging...this is compounded here by the absence of a matched normal control. Similarly, a fraction of WES/WGS samples lacked a germline control. 75% of cases have both RNA-seq and WGS/WES data. This could provide some level of cross- platform validation, but that was not described. The manuscript does not appear to report novel somatic variants which tempers this concern somewhat. Still, some orthogonal validation of calls made only by RNA-seq appears warranted. This could be addressed by sequencing remission samples or matched normal tissue, if available.

We appreciate the technical challenge of SNV/indel calling from RNA-Seq data which stems from various reasons including 1. Lack of germline controls 2. Sequencing errors or nonsense-mediated decay 3. Low expression of the transcript. We focused on 87 genes recurrently altered in myeloid neoplasms and utilized an indel caller optimized for RNA-seq BAM files^{5,6}. along with multiple layers of filtering of frequent SNPs to reduce false positive calls as stated in the method section.

We extensively validated this approach in our previous study utilizing RNA-seq based mutation calling pipeline⁷ and showed that 97.8% (45/46) of mutation calls from RNA-seq BAM files validated by target capture sequencing (TCS) as orthogonal validation. In this study, we cross-validated mutation calls from RNA-seq data using DNA calls (tumor/germline-matched WGS/WES, Fig.S1E), validating 97.3% (364/374) of the RNA-seq. Among the unvalidated 10 calls, 3 calls are likely due to low VAF (8.2~11.2%)

and low coverage in WGS data as the variants were detected in the DNA upon manual inspection. We did not find any supporting reads in four calls with low VAF (2.1~8.5%). Two calls were found to be germline mutations, which are Benign or Variants of Unknown Significance (VUS), which did not affect the downstream analysis. This is further evidence to support the effectiveness of our somatic mutation calling from RNA-Seq workflow with few germline variants after filtering. One *CEBPA* mutation (P189_H191>H) with high VAF (27.5%) in RNA-seq data was not found in DNA.

5. Conversely, the lack of matched normal sequencing data from many patients could have led to an underestimate of the number of cases with inherited predisposition drivers. Indeed, the frequency reported here is lower than expected.

Germline mutation calls could be done in cases with paired DNA data (70.9%), and we focused on 15 genes with known predisposition defined in the WHO classification for germline mutation calls. This limitation should have underestimated the total frequency of germline predispositions compared with previous publications (e.g., pediatric cohort⁸ or adult and pediatric cohort⁹). However, it is also notable that cases with germline predispositions show similar expression profile with somatic mutations (*CEBPA*) or defined by somatic gene alterations.

WHO (5th) Category	Molecular category
Myeloid neoplasms with germline CEBPA variant	CEBPA
Myeloid neoplasms with germline CEBPA variant	CEBPA
Myeloid neoplasms with germline CEBPA variant	CEBPA
Myeloid neoplasms with germline ETV6 variant	KMT2Ar
Myeloid neoplasms with germline GATA2 variant	Unclassified
Myeloid neoplasms with germline GATA2 variant	NPM1
Myeloid neoplasms with germline RUNX1 variant	KMT2A-PTD
Myeloid proliferation associated with Down syndrome	GATA1
Myeloid proliferation associated with Down syndrome	GATA1
Myeloid proliferation associated with Down syndrome	GATA1

6. The organization of categories in Fig 2D gives the false impression that many patients in genetically-defined subgroups from WHO are reassigned to new categories in the proposed molecular framework, whereas this is just a consequence of the way the categories are ordered. *NPM1*, *NPU98r*, and *CEBPA* should be presented in the same order in both systems. This will fix the problem and allow the new entities to stand out (e.g., *UBTF*, *GLIS4*, *GATA1*). The last sentence in this section on p8 should emphasize that with the identification of 12 new molecular categories, the new classification system captures 91.4% of cases, up from 68.3% by WHO.

As stated in an answer to comment 8 from Reviewer #2, we ordered the categories according to the

frequencies in WHO classification and molecular categories respectively. We also changed the last phasing of paragraph 2 to emphasize the better coverage of our molecular categories compared with the WHO classification.

7. The largest fraction of cases assigned to the Unclassified category in the molecular system were classified as myelodysplasia-related in WHO. This should be noted in the section on Unclassified cases (and perhaps retained as a non-molecularly defined category since they have distinct clinical features). In adults, many of these cases would harbor splicing gene mutations. These mutations are less common in pAML, although it is worth noting that one of the most frequently mutated splicing genes (SRSF2) was not included in the 86 gene panel.

We thank the reviewer for raising this discussion over acute myeloid leukemia, myelodysplasia-related (AML-MR) category in the WHO classification in the context of pAML. We also included *SRSF2* in the gene panel (87 genes) in the revised manuscript but did not identify any additional pathogenic/likely pathogenic mutations in *SRSF2* genes, in addition to one mutation already detected by WGS and included in the original manuscript.

In Fig.5 and Fig.S8, we focused more on AML-MR to see whether AML-MR cases have unique expression profiles. Compared with *ETV6* or *RUNX1* alteration predominantly found in immature or T-ALL like clusters, AML-MR defining alterations are found in multiple clusters such as *HOXA* cluster together with *KMT2A* or *HOXB* cluster with *NPM1* or *UBTF*. Considering the differential frequency of splicing mutations between pediatric AML and adult AML, and rarity of pediatric AML, we suppose that pediatric AML with myelodysplasia-related alterations does not necessarily represent a unique molecular category or similar disease entity with adult AML-MR, and further investigation will be required to understand factors contributing to the expression and disease phenotypes.

8. The clinical experience with a single Menin inhibitor is mentioned in the discussion. To avoid the perception of bias, this discussion should be broadened slightly since there are now several compounds in the clinic with preliminary results reported.

We appreciate this comment because we did not intend to emphasize a specific Menin inhibitor to be a promising drug, while there is only one publication reporting clinical outcomes from these 6 clinical trials investigating Menin inhibitors in AML that we could find. These clinical studies are included as references 100-106 in the revised manuscript.

100 Issa, G. C. *et al.* The menin inhibitor revumenib in KMT2A-rearranged or NPM1-mutant leukaemia. *Nature* **615**, 920-924, doi:10.1038/s41586-023-05812-3 (2023).

101 Sumitomo Pharma Oncology, I. *A Study of DSP-5336 in Relapsed/Refractory AML/ ALL With or Without MLL Rearrangement or NPM1 Mutation*, <<https://classic.clinicaltrials.gov/show/NCT04988555>> (2025).

102 Biomea Fusion, I. *Study of BMF-219, a Covalent Menin Inhibitor, in Adult Patients With AML, ALL (With KMT2A/ MLL1r, NPM1 Mutations), DLBCL, MM, and CLL/SLL*, <<https://classic.clinicaltrials.gov/show/NCT05153330>> (2024).

103 Center, M. D. A. C. *DS-1594b With or Without Azacitidine, Venetoclax, or Mini-HCVD for the Treatment of Relapsed or Refractory Acute Myeloid Leukemia or Acute Lymphoblastic Leukemia*,

- <<https://classic.clinicaltrials.gov/show/NCT04752163>> (2024).
 104 Syndax, P. *A Study of SNDX-5613 in R/R Leukemias Including Those With an MLLr/KMT2A Gene Rearrangement or NPM1 Mutation*,
 <<https://classic.clinicaltrials.gov/show/NCT04065399>> (2025).
 105 Kura Oncology, I. *First in Human Study of Ziftomenib in Relapsed or Refractory Acute Myeloid Leukemia*,
 <<https://classic.clinicaltrials.gov/show/NCT04067336>> (2024).
 106 Janssen, R. & Development, L. L. C. *A Study of JNJ-75276617 in Participants With Acute Leukemia*,
 <<https://classic.clinicaltrials.gov/show/NCT04811560>> (2023).

Minor:

There are numerous callouts to figure panels that are discordant, presumably because text was not updated after figures were changed.

We have updated the callout according to the revision of figures and double-checked the concordance.

- 1 Martelli, M. P. *et al.* Novel NPM1 exon 5 mutations and gene fusions leading to aberrant cytoplasmic nucleophosmin in AML. *Blood* **138**, 2696-2701, doi:10.1182/blood.2021012732 (2021).
- 2 Harrell, F. E., Jr., Califf, R. M., Pryor, D. B., Lee, K. L. & Rosati, R. A. Evaluating the yield of medical tests. *JAMA* **247**, 2543-2546 (1982).
- 3 Pollard, J. A. *et al.* Sorafenib in Combination With Standard Chemotherapy for Children With High Allelic Ratio FLT3/ITD+ Acute Myeloid Leukemia: A Report From the Children's Oncology Group Protocol AAML1031. *J Clin Oncol* **40**, 2023-2035, doi:10.1200/JCO.21.01612 (2022).
- 4 Lee, S. H. R. *et al.* Association of Genetic Ancestry With the Molecular Subtypes and Prognosis of Childhood Acute Lymphoblastic Leukemia. *JAMA Oncol* **8**, 354-363, doi:10.1001/jamaoncol.2021.6826 (2022).
- 5 Hagiwara, K. *et al.* RNAIndel: discovering somatic coding indels from tumor RNA-Seq data. *Bioinformatics* **36**, 1382-1390, doi:10.1093/bioinformatics/btz753 (2020).
- 6 Hagiwara, K., Edmonson, M. N., Wheeler, D. A. & Zhang, J. indelPost: harmonizing ambiguities in simple and complex indel alignments. *Bioinformatics*, doi:10.1093/bioinformatics/btab601 (2021).
- 7 Umeda, M. *et al.* Integrated Genomic Analysis Identifies UBTF Tandem Duplications as a Recurrent Lesion in Pediatric Acute Myeloid Leukemia. *Blood Cancer Discov* **3**, 194-207, doi:10.1158/2643-3230.BCD-21-0160 (2022).
- 8 Samaraweera, S. E. *et al.* Childhood acute myeloid leukemia shows a high level of germline predisposition. *Blood* **138**, 2293-2298, doi:10.1182/blood.2021012666 (2021).
- 9 Kim, B. *et al.* Prevalence and clinical implications of germline predisposition gene mutations in patients with acute myeloid leukemia. *Sci Rep* **10**, 14297, doi:10.1038/s41598-020-71386-z (2020).

Decision Letter, first revision:

19th Sep 2023

Dear Dr Klco,

Thank you for submitting your revised manuscript "Proposal of a new genomic framework for categorization of pediatric acute myeloid leukemia associated with prognosis" (NG-A62581R). It has now been seen by the original referees and their comments are below. The reviewers find that the paper has improved in revision, and therefore we'll be happy in principle to publish it in Nature Genetics, pending minor revisions to satisfy our editorial and formatting guidelines.

Sincerely,

Safia Danovi
Editor
Nature Genetics

Reviewer #1 (Remarks to the Author):

The authors have fully answered my questions.

Reviewer #2 (Remarks to the Author):

I thank the authors for clear and comprehensive responses to my comments. I have no further inquiries. This is a manuscript that will be valuable to the field of pediatric AML.

Sincerely,
Rachel Rau
Seattle Children's Hospital

Reviewer #3 (Remarks to the Author):

No additional comments/criticisms.

Final Decision Letter:

5th Dec 2023

Dear Dr Klco,

I am delighted to say that your manuscript "A new genomic framework to categorize pediatric acute myeloid leukemia" has been accepted for publication in an upcoming issue of Nature Genetics.

Your paper will be published online after we receive your corrections and will appear in print in the next available issue. You can find out your date of online publication by contacting the Nature Press Office (press@nature.com) after sending your e-proof corrections. Now is the time to inform your Public Relations or Press Office about your paper, as they might be interested in promoting its publication. This will allow them time to prepare an accurate and satisfactory press release. Include your manuscript tracking number (NG-A62581R1) and the name of the journal, which they will need when they contact our Press Office.

Please note that Nature Genetics is a Transformative Journal (TJ). Authors may publish their research with us through the traditional subscription access route or make their paper immediately open access through payment of an article-processing charge (APC). Authors will not be required to

make a final decision about access to their article until it has been accepted. [Find out more about Transformative Journals](https://www.springernature.com/gp/open-research/transformative-journals)

Authors may need to take specific actions to achieve [compliance with funder and institutional open access mandates](https://www.springernature.com/gp/open-research/funding/policy-compliance-faqs). If your research is supported by a funder that requires immediate open access (e.g. according to [Plan S principles](https://www.springernature.com/gp/open-research/plan-s-compliance)) then you should select the gold OA route, and we will direct you to the compliant route where possible. For authors selecting the subscription publication route, the journal's standard licensing terms will need to be accepted, including [those licensing terms](https://www.nature.com/nature-portfolio/editorial-policies/self-archiving-and-license-to-publish) will supersede any other terms that the author or any third party may assert apply to any version of the manuscript.

If you have not already done so, we invite you to upload the step-by-step protocols used in this manuscript to the Protocols Exchange, part of our on-line web resource, natureprotocols.com. If you complete the upload by the time you receive your manuscript proofs, we can insert links in your article that lead directly to the protocol details. Your protocol will be made freely available upon publication of your paper. By participating in natureprotocols.com, you are enabling researchers to more readily reproduce or adapt the methodology you use. [Natureprotocols.com](http://natureprotocols.com) is fully searchable, providing your protocols and paper with increased utility and visibility. Please submit your protocol to <https://protocolexchange.researchsquare.com/>. After entering your nature.com username and password you will need to enter your manuscript number (NG-A62581R1). Further information can be

found at <https://www.nature.com/nature-portfolio/editorial-policies/reporting-standards#protocols>

Sincerely,

Safia Danovi
Editor
Nature Genetics